# Lineage tracing identifies heterogeneous hepatoblast contribution to cell lineages and postembryonic organ growth dynamics

Iris. A. Unterweger[1,2], Julie Klepstad[3,4], Edouard Hannezo[5], Pia R. Lundegaard[2], Ala Trusina[3], Elke A. Ober[1,2]*

1 University of Copenhagen, NNF Center for Stem Cell Biology (DanStem), Copenhagen N, Denmark, 2 University of Copenhagen, Department of Biomedical Sciences, Copenhagen N, Denmark, 3 Niels Bohr Institute, University of Copenhagen, Copenhagen, Denmark, 4 Andalusian Center for Developmental Biology, CSIC, University Pablo de Olavide, Seville, Spain, 5 Institute of Science and Technology, Klosterneuburg, Austria

* elke.ober@sund.ku.dk

**Data Availability Statement:** All relevant data are within the paper and its Supporting Information files. The code can be accessed via the gitbub

## Abstract

To meet the physiological demands of the body, organs need to establish a functional tissue architecture and adequate size as the embryo develops to adulthood. In the liver, uni- and bipotent progenitor differentiation into hepatocytes and biliary epithelial cells (BECs), and their relative proportions, comprise the functional architecture. Yet, the contribution of individual liver progenitors at the organ level to both fates, and their specific proportion, is unresolved. Combining mathematical modelling with organ-wide, multispectral FRaeppli-NLS lineage tracing in zebrafish, we demonstrate that a precise BEC-to-hepatocyte ratio is established (i) fast, (ii) solely by heterogeneous lineage decisions from uni- and bipotent progenitors, and (iii) independent of subsequent cell type–specific proliferation. Extending lineage tracing to adulthood determined that embryonic cells undergo spatially heterogeneous three-dimensional growth associated with distinct environments. Strikingly, giant clusters comprising almost half a ventral lobe suggest lobe-specific dominant-like growth behaviours. We show substantial hepatocyte polyploidy in juveniles representing another hallmark of postembryonic liver growth. Our findings uncover heterogeneous progenitor contributions to tissue architecture-defining cell type proportions and postembryonic organ growth as key mechanisms forming the adult liver.

## Introduction

Liver formation requires the timely differentiation of multipotent progenitor cells into specific cell types that form the building blocks of the organ. Relative proportions of these cell types are critical for establishing a specialised tissue architecture mediating physiologic liver functions. During embryonic and postembryonic growth, the liver increases in size to meet the growing physiological demands. Yet, how individual progenitors contribute to distinct cell lineages and subsequent growth are fundamental questions in organogenesis.

repository https://github.com/JulieKlepstad/LiverDevelopment.

**Funding:** This work is supported by Novo Nordisk Foundation grant NNF17CC0027852 (EAO); Nordisk Foundation grant NNF19OC0058327 (EAO); Novo Nordisk Foundation grant NNF17OC0031204 (PRL); https://novonordiskfonden.dk/en/; Danish National Research Foundation grant DNRF116 (EAO and AT); https://dg.dk/en/; John and Birthe Meyer Foundation (PRL) and European Research Council (ERC) under the EU Horizon 2020 research and Innovation Programme Grant Agreement No. 851288 (EH); https://research-and-innovation.ec.europa.eu/funding/funding-opportunities/funding-programmes-and-open-calls/horizon-2020_en. The funders had no role in study design, data collection and analysis, decision to publish, or preparation of the manuscript.

**Competing interests:** The authors have declared that no competing interests exist.

**Abbreviations:** BABB, benzyl alcohol–benzyl benzoate; BEC, biliary epithelial cell; DAPI, 4′,6-diamidino-2-phenylindole; DMSO, dimethylsulfoxide; dpf, day postfertilization; EdU, 5-ethynyl-2′-deoxyuridine; FP, fluorescent protein; hpf, hours post fertilization; PTU, 1-phenyl 2-thiourea; ROI, region of interest; SL, standard length.

The liver consists mostly of hepatocytes and biliary epithelial cells (BECs), also called cholangiocytes, which, together with mesenchymal cell types, are arranged in a characteristic architecture executing essential liver functions. On the tissue scale, the mammalian liver lobes are divided into liver subunits with a central vein and portal triads consisting of portal veins, arteries, and biliary ducts at the edges, and hepatocytes distribute within the subunit along sinusoids connecting the main blood vessels [1]. In zebrafish, the lobes are not subdivided; instead, the central vein resides in the core of each lobe, and the portal veins at the periphery [2,3]. Hepatocytes align along sinusoids between the 2 veins. Both sinusoids and the intrahepatic bile ductules are organised throughout the lobe in complementary mesh-like networks [4].

During development, hepatic progenitors, called hepatoblasts, are specified in the ventral foregut endoderm by signals from the adjacent mesoderm [5]. Immunohistochemistry studies of the rat liver initially suggested the bipotent nature of hepatoblasts, the ability to differentiate into both BECs and hepatocytes [6]. Bipotency was subsequently demonstrated in vitro by culturing mouse hepatoblasts isolated by selected surface markers in respective culture media [7,8] and more recently in organoids [9]. Lineage tracing of early definitive foregut endoderm in mice, labelled at E7.75, showed contribution to both lineages pointing to bipotency. However, recombination was induced prior to liver specification [10]. Instead, tracing of Lgr5[+] hepatoblasts from E9.5, representing 2% of hepatoblasts at this time point, showed that uni- and bipotent hepatoblasts contribute solely hepatocytes or hepatocytes and BECs when focussing on the portal triad [9]. Yet, a systematic organ-wide understanding of uni- and bipotent lineage decisions is missing. Overall, these studies indicate a gradual restriction of progenitor potential over time. In line with transcriptional profiling in mice suggesting the transition from hepatoblasts to hepatocytes occurs by default in the absence of specific inductive signals, whereas the hepatoblast to BEC transition represents a regulated process [11–13]. Moreover, whether a heterogeneous hepatoblast potential represents a conserved strategy across vertebrate liver formation and how the precise cell type proportions critical for a functional organ architecture are established are open questions.

Once the nascent tissue organisation is established, the liver transitions into a growth phase [13]. This occurs in zebrafish around 5 days postfertilization (dpf), when the liver consists of 2 lobes, the left and right lobe, and takes up organ-specific functions [14]. As the liver enlarges during postembryonic growth, a third liver lobe, the ventral lobe, arises [15]. While the molecular mechanisms of liver cell type differentiation are gradually being elucidated [4], postnatal growth and the transition to the adult organ remain generally poorly understood [16,17]. To accommodate the 900-fold increase in cell number between 5 dpf and 1.5 years, each embryonic liver cell in zebrafish divides theoretically 10 times [18]. Lineage tracing over a similar period revealed that new hepatocytes arise exclusively from the proliferation of existing hepatocytes [18]. However, BECs can also transdifferentiate into hepatocytes and contribute to the hepatocyte pool in homeostasis [19]. Similar differences have been seen in mice, attributed mostly to diverse lineage tracing approaches [20]. Furthermore, the contribution of individual hepatoblasts to the growth of the adult liver remains unknown: for example, do all hepatoblasts produce an equal number of progeny or do some generate more than others [21,22]? Mechanistically, this could be controlled spatially, for example, by growth zones at the organ periphery during development [23–25] or regionally within the lobule [26]. Overall, there is a large gap in our understanding of postembryonic liver growth across all liver lobes, as well as between species.

Combining lineage tracing with whole-mount imaging and mathematical modelling in zebrafish, we here show that heterogeneous lineage contributions of progenitors are sufficient for establishing the precise proportion of BECs and hepatocytes comprising the functional liver architecture. Furthermore, by morphological and clonal studies, we demonstrate that

embryonic cells contribute heterogeneously to postembryonic liver growth, including giant clusters driving the distinct growth behaviour of the ventral lobe during metamorphosis.

## Results

### Mathematical modelling for establishing a precise BEC to hepatocyte proportion

To elucidate how the correct number and proportion of hepatocytes and BECs arise during liver development (Fig 1A), we first determined their cell numbers at 5 dpf when hepatic tissue organisation is established, and the liver takes up function (Fig 1B). Visualising hepatocytes and BECs by expression of Hnf4α [27] and the Notch reporter *tp1:EGFP* [28], respectively, we showed that hepatocytes outnumber BECs by 9-fold (Fig 1C). Asking how this 1:9 BEC-to-hepatocyte ratio is established, we turned to mathematical modelling considering 2 parameters: lineage potential of the initial hepatoblast population and proliferation rates of the differentiated cell types (Figs 1D–1F and S1A–S1C). The simplest approach to achieve this 1:9 ratio would solely comprise 2 types of unipotent hepatoblasts, one consistently contributes to the BEC lineage (10%) and the other one generates hepatocytes (90%). However, since mammalian liver formation involves bipotent hepatoblasts [9], they were included in all our models.

First, a model with 100% bipotent hepatocytes results in a 1:1 BEC-to-hepatocyte ratio (model 1; Fig 1D), we decreased in model 2 the division rate of BECs by an estimated 50%. This scenario generated a 1:2.3 BEC-to-hepatocyte ratio, suggesting that moderately different cell division times alone are not sufficient to produce the in vivo proportion (model 2; Fig 1E). Next, varying the composition of hepatoblasts by introducing uni- and bipotent potentials lead to ratios with a higher hepatocyte fraction (models 3 to 6; Figs 1F and S1A–S1C). The model producing an outcome (ratio 1:8.5) closest to the 1:9 in vivo ratio contains 80% unipotent hepatocyte-producing hepatoblasts and 20% bipotent ones, exhibiting equal proliferation rates (model 6; Fig 1F). A recent single cell transcriptome study in mice predicted unipotent hepatoblasts producing only BECs [9]. Adding a low probability of 5% unipotent hepatoblasts contributing solely BECs resulted in a 1:5.7 (model 4; S1B Fig) compared to the 1:8.5 ratio in model 6 (Fig 1F), whereas decreasing BEC proliferation rates in this model similar to above established a 1:11.8 ratio over time (model 5; S1C Fig).

It is noteworthy that in the various models, respective BEC-to-hepatocyte ratios are established with different velocities. Specifically, a BEC–hepatocyte equilibrium is reached faster in models 4 and 6 with the greatest hepatoblast heterogeneity, while solely altering proliferation rates between cell types in model 2 takes about 3 times longer.

In summary, mathematical modelling predicts that the in vivo 1:9 BEC-to-hepatocyte ratio cannot be established by decreasing the BEC proliferation rate alone. In contrast, a heterogeneous progenitor lineage potential is sufficient to achieve such proportions.

### BEC and hepatocyte proliferation dynamics during embryonic development

Based on mathematical models 2 and 5, unequal proliferation rates can contribute to a differential BEC-to-hepatocyte ratio. To test whether in vivo division rates between BECs and hepatocytes are similar or differ, we examined cell proliferation by 5-ethynyl-2′-deoxyuridine (EdU) incorporation between 48 and 144 hours post fertilization (hpf). Starting at 48 hpf, 16% to 18% of both cell types proliferate, at a rate continuously decreasing until 120 hpf, when only 2% to 4% are EdU positive (Fig 2A and 2B). Although higher EdU incorporation in BECs between 96 and 120 hpf (Fig 2B) indicates transiently higher BEC proliferation, the overall 1:9

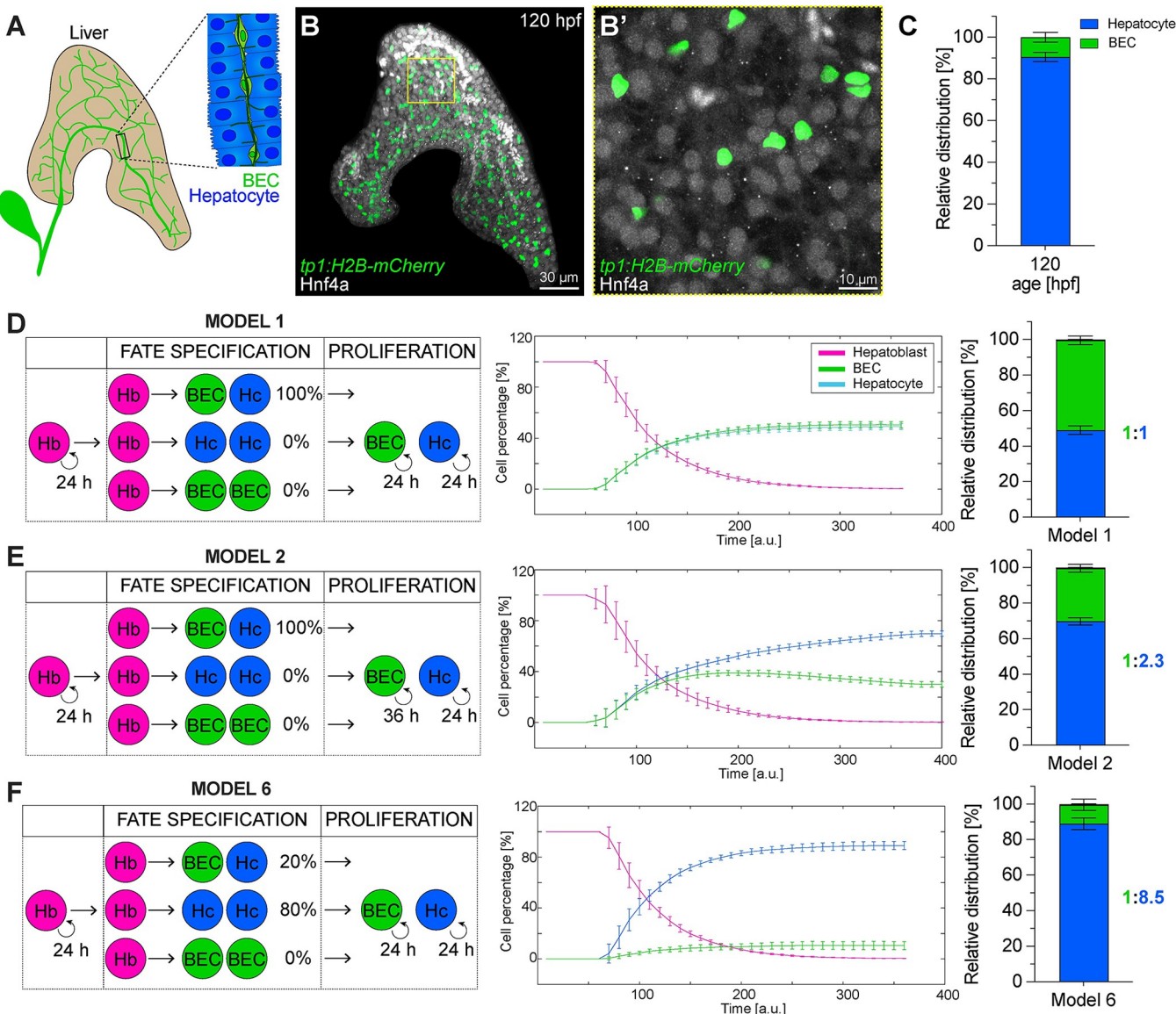

**Fig 1. Establishment of BEC and hepatocyte lineages: in vivo cell type quantification and in silico modelling.** (**A**) Schematic of a 5-dpf liver, highlighting the biliary network. (**B-B'**) Maximum projection (200 μm z-stack) of a 120-hpf liver expressing *tp1:H2B-mCherry* (BEC) and stained for Hnf4α (hepatocyte). Autofluorescent blood cells appear in bright white. (*N* = 4, *n* ≥ 12 livers) (**C**) Relative distribution of BECs and hepatocytes at 120 hpf (*N* = 4, *n* ≥ 12 livers). (**D-F**) Mathematical models simulating hepatoblast differentiation employing different parameter combinations: proliferation rates of differentiated cell types is equal (**D, F**) or slower in BECs (**E**). Hepatoblasts either are all bipotent (**D, E**) or represent a heterogeneous population with mixed probabilities for uni-or bipotent differentiation (**F**). Plots showing the simulated cell proportions over simulation time (*n* = 10) and the final cell type ratio in bar graphs. The numerical values that were used to generate the graphs in (**C-F**) can be found in S1 Data. BEC, biliary epithelial cell; dpf, day postfertilization; Hb, hepatoblast; Hc, hepatocyte; hpf, hours post fertilization.

BEC-to-hepatocyte ratio remains unchanged, suggesting a balancing mechanism to sustain a stable tissue organisation. Both the total number of BECs and hepatocytes increases by 8.4-fold between 48 and 120 hpf (Fig 2C and 2D), while the liver volume increases disproportionately by 20-fold within the same timeframe (Fig 2E).

Our results indicate progenitor potential as a major factor for the establishment of the 1:9 BEC-to-hepatocyte ratio in vivo, mirrored in mathematical models 4 and 6 (Figs 1F and S1B).

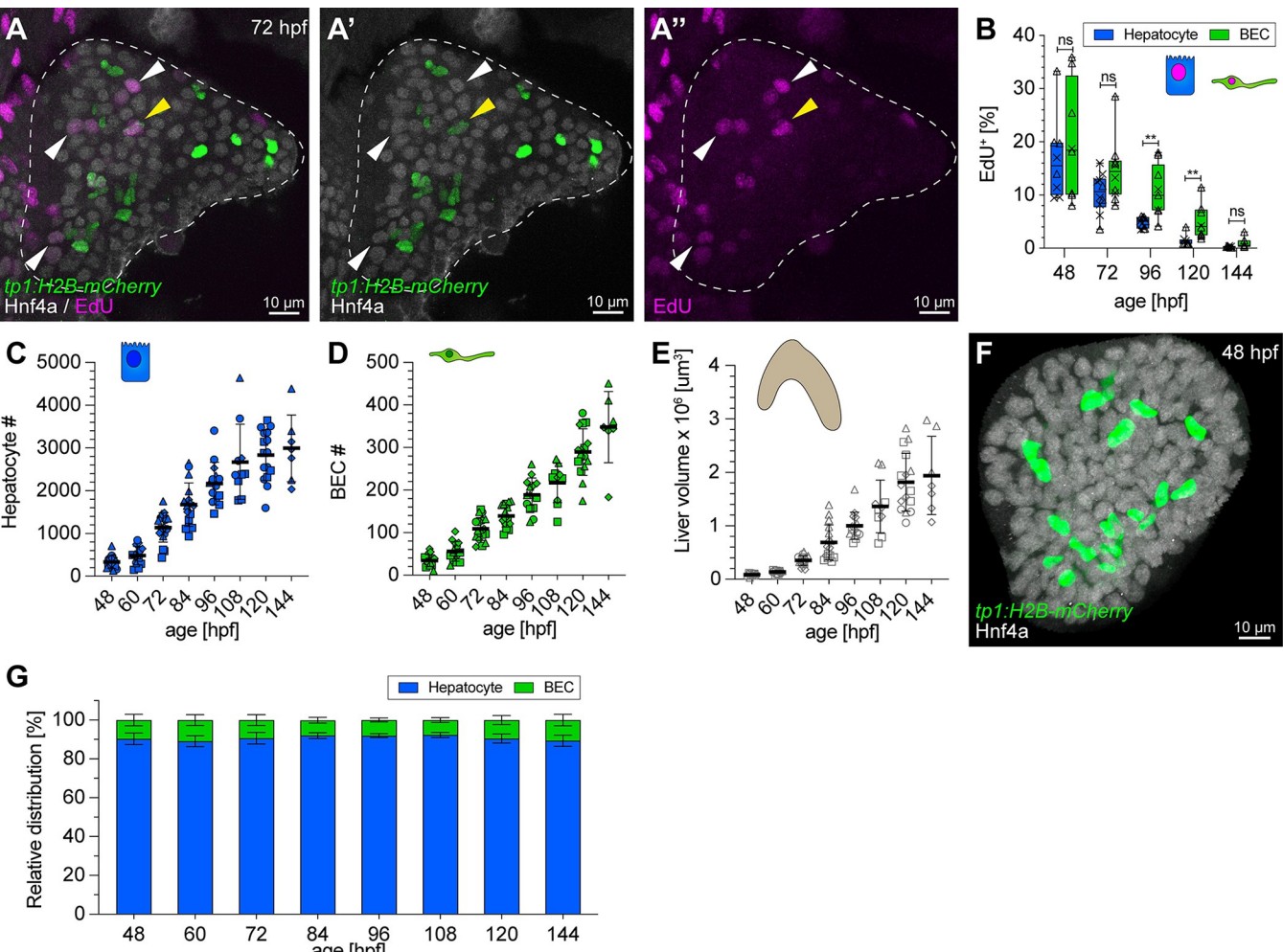

**Fig 2. Hepatic proliferation dynamics and early establishment of a 1:9 BEC:hepatocyte ratio during embryonic development.** (**A**) Approximately 5 μm projection of a 72-hpf liver expressing *tp1:H2B-mCherry* (BEC), stained for Hnf4a (hepatocytes) and EdU (proliferating cells). Yellow and white arrowheads highlight proliferating BECs and hepatocytes, respectively (*N* = 2, *n* = 10 livers). (**B**) Graph showing the proportion of EdU⁺ proliferating hepatocytes and BECs over time (*N* = 2, *n* ≥ 8). (**C, D**) Graph showing hepatocyte (**C**) and BEC (**D**) cell numbers during development (*N* = 4, *n* ≥ 12 livers). (**E**) Quantification of total liver volume during development determined in embryos in BABB (*N* = 4, *n* ≥ 12 livers). (**F**) Maximum projection (20 μm z-stacks) of a 48-hpf liver expressing *tp1:H2B-mCherry* (BEC) and stained for Hnf4α (hepatocyte). (**G**) Relative distribution of BECs and hepatocytes during development from 48 to 144 hpf (*N* = 4, *n* ≥ 12 livers). (**B-E**) Different shape data points indicate different experiments. The numerical values that were used to generate the graphs in (**B-E**, **G**) can be found in S1 Data. BEC, biliary epithelial cell; EdU, 5-ethynyl-2′-deoxyuridine; hpf, hours post fertilization.

## The mature BEC to hepatocyte ratio is already established early in liver development

Another distinguishing hallmark between the different models is the velocity, up to 3-fold different, by which the BEC-to-hepatocyte ratio arises. We determined cell type numbers throughout development to assess when the 1:9 BEC-to-hepatocyte ratio is established after cell type specification in vivo. Unexpectedly, the 1:9 ratio of BECs to hepatocytes is already reached at 48 hpf (Fig 2F and 2G) only a few hours after the onset of BEC differentiation (Fig 2F). This finding together with similar BEC and hepatocyte proliferation rates (Fig 2B) strongly supports models 4 and 6, in which hepatoblast heterogeneity is sufficient to rapidly establish the final ratio and maintain it over time.

Asking whether this distinct cell type ratio is characteristic for the differentiating embryonic liver or important for tissue functionality, we next examined in vivo cell numbers in postembryonic livers (S1D–S1G Fig). We determined a 1:7.9 BEC-to-hepatocyte ratio in juvenile and an average 1:8.75 in adult livers (periphery: 1:6.4; and centre:1:11.1), suggesting that similar cell type proportions are maintained from the embryonic to the mature liver.

## Lineage tracing identifies uni- and bipotent hepatoblasts in vivo

Next, to investigate hepatoblast potential and its role in establishing cell type proportions, we applied unbiased lineage tracing strategies. Hepatoblasts are specified around 23 hpf [4,29–31], and expression of the Notch reporter *tp1:EGFP* visualises the first BECs at 45 hpf [28]. In parallel, hepatocyte differentiation begins between 40 and 60 hpf [4]. For hepatoblast lineage tracing, we used the multicolour labelling system FRaeppli-NLS, which, upon conditional activation, stochastically labels nuclei with one of 4 fluorescent proteins (FPs): TagBFP, mTFP1, E2-Orange, or mKate2 [32] (Figs 3A and S3A). The spectra of the FRaeppli FPs are distinct from the EGFP spectra allowing for the simultaneous use with transgenic *tp1:EGFP* expressed in BECs. To ensure that the *UAS:fraeppli-nls* transgene (short: *fraeppli-nls*) is expressed in progenitors and maintained in hepatocytes and BECs, we used *prox1a:kalTA4* as a KalTA4-driver (Figs 3A and S3A), since Prox1 is expressed in hepatoblasts and differentiated hepatocytes and BECs (S2A–S2C Fig) [33].

Hepatoblast labelling was achieved by PhiC31 integrase mediated recombination of the *fraeppli-nls* colour cassette (Figs 3A and S3A). Conditional recombination of the FRaeppli-cassette in individual cells prior to cell differentiation in *fraeppli-nls* was achieved with *hsp70l:phiC31* and induced by heat shock at 26 hpf (Fig 3B and 3C). PhiC31 maturation and subsequent attB/attP recombination, initiating the stable expression of one of 4 FPs per cell, takes about 6 to 7 hours (Figs 3B, 3C and S2D–S2F) [32] and thus initiates prior to fate commitment. Based on total liver cell numbers and cell doubling times, we estimated that FRaeppli-labelled hepatoblasts would undergo maximally one cell division before lineage decision at around 40 to 42 hpf (S2G Fig). For clonal analysis, we fixed embryos at 100 hpf, when all 4 FRaeppli FPs are strongly expressed (S2H, S2I, S2K and S2L Fig). To determine cell fates within a clone, we combined *fraeppli-nls; prox1a:kalTA4; hsp70l:phiC31* with *tp1:EGFP* to distinguish BEC from hepatocyte fate. In addition, samples without *tp1:EGFP* expression were included using nuclear shape as an indicator of cell fate, given that at 100 hpf, BEC nuclei are mostly elongated, while hepatocyte nuclei are round (S2A–S2C Fig) [34]. Given that cell rearrangement can lead to fragmentation and merging of clones [32,35], we established rigorous rules to define clones. Considering both the range of cell movement determined by live imaging and the volume increase of the entire organ (S2G and S2J Fig), clones were defined as labelled cells of the same colour located within a 70-μm radius (S3B Fig).

With this strategy, 214 clones were analysed for their lineage contribution to hepatocytes and BECs. A recombined clone was found in 1.8% of the control samples, indicating low non-specific recombination. Interestingly, out of all heat shock–induced embryos, 78% exclusively labelled hepatocytes, while the remaining 22% displayed clones also containing labelled BECs. Overall, we observed 3 distinct clonal outcomes: first, mixed clones accounting for 10.7% of cases and indicated to originate from bipotent hepatoblasts (Figs 3D and S3C). Second, the majority of clones, 88.8%, consisted of hepatocytes only, in line with a unipotent hepatoblast potential (Figs 3E, S3D and S3E). Finally, in 0.5% of clones, all labelled cells were BECs, suggesting unipotent hepatoblasts with a BEC-restricted lineage contribution (Figs 3F, S3F and S3G). Mixed clones were detected mostly in densely labelled livers, in line with increasing labelling frequency (S4A Fig). Lineage tracing of stochastically labelled hepatoblasts reveals

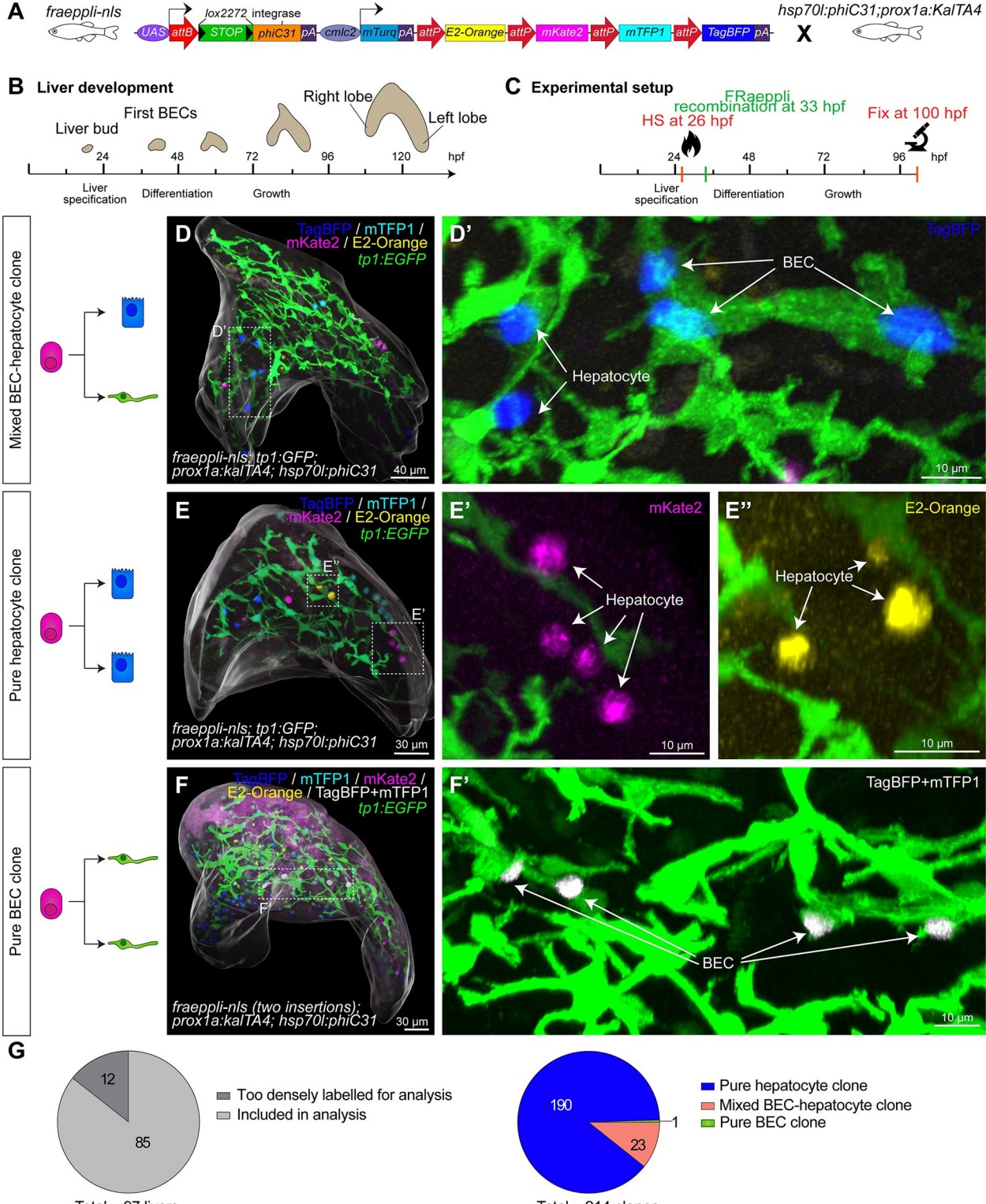

**Fig 3. Quantitative lineage tracing identifies uni- and bipotent hepatoblast contributions during lineage decisions.** (**A**) Schematic of FRaeppli-NLS cassette including attB and attP sites for PhiC31-mediated recombination and the 4 FRaeppli FPs: TagBFP, mTFP1, mKate2, and E2-Orange. Recombination is induced by combining *fraeppli-nls* with *hsp70l:phiC31; prox1a:kalTA4*; see S3A Fig. (**B**) Key steps of liver development in zebrafish: After hepatoblast specification, the differentiation into BECs and hepatocytes is initiated at around 42 hpf. Differentiated cells acquire polarity and form a functional architecture by 120 hpf. (**C**) Experimental strategy for tracing progeny of individual hepatoblasts using *fraeppli-nls*:

Heat shock at 26 hpf controls PhiC31 expression followed by attB-attP recombination. Embryos were fixed at 100 hpf for analysis. (**D-F**) Whole-mount livers at 100 hpf showing (**D**) mixed clone composed of hepatocytes and BECs (D') ($N = 6$, $n = 23$ clones); (**E**) clones formed by pure hepatocytes (E'-E'') ($N = 6$, $n = 190$ clones); and (**F**) example of pure BEC clone coexpressing TagBFP and mTFP1 (white, coexpressing cells were manually segmented and masked). (F') ($N = 2$, $n = 2$ clones). (**D-F**) An overall segmentation of the whole liver tissue is shown in transparent grey. (**G**) Pie charts showing the total number of labelled embryos and clones with manually assigned lineage contributions ($N = 6$, $n = 214$ clones; in 2 of the 6 experiments, nuclear shape indicated BEC fate). The numerical values that were used to generate the graphs in (**G**) can be found in S1 Data. BEC, biliary epithelial cell; FP, fluorescent protein; hpf, hours post fertilization.

their heterogeneous potential, encompassing bipotent hepatoblasts, as well as a high fraction of unipotent hepatoblasts contributing predominantly to the hepatocyte lineage (Fig 3G), as predicted by mathematical model 6 (Fig 1F).

## Hepatoblasts contribute stochastically to embryonic liver growth

Hepatoblast lineage tracing revealed a striking range of clone sizes, raising the question as to whether their proliferative capacity and contribution to the overall liver is controlled at the level of the individual progenitor or stochastic. Clone size of pure hepatocyte clones varied, from 1 to 33 cells (Fig 4A and 4C–4E), independent of clone colour (Fig 4A). Mixed BEC–hepatocyte clones were in the same size range (S4B Fig) and with varying cell type distribution (S4C Fig). Given the range in clone size, we converted each cell number into the number of cell divisions each progenitor had undergone. This showed that the largest clone arose from a single progenitor that underwent 6 divisions, the majority, 22.1%, divided once, while a substantial number of hepatoblasts, 7.9%, did not divide (Fig 4B). Comparing this clonal division range to a Poisson distribution, which is the simplest null model where the distribution arises simply from the stochasticity of division timing of a single homogeneous population. The good fit between data and the simple model, in the absence of any free parameter apart from the average division time, suggests a fully stochastic proliferation behaviour (S4D and S4E Fig).

To rule out a bias arising from manual clone assignment, we defined clonality mathematically considering all labelled cells per liver (S4F Fig). We first calculated the probability that a given cell had a neighbour with the same or a different colour (S4G Fig) [36,37]. Then, looking at subsets of livers with a defined number of total labelled cells revealed that clonality assignment becomes unprecise when a threshold of 40 to 50 labelled cells per liver is exceeded (S4G Fig). This also determined that clones of different colours are usually at least 50 μm apart, suggesting that labelled cells more than 50 μm apart should not be considered clones. Next, we compared the size distribution of manually defined clones to theoretically reconstructed clones (S4H Fig). This analysis revealed that including samples with less than 40 labelled cells and grouping cells within a 45-μm distance between clones represents a suitable ruleset. Interestingly, the reconstructed clone size distribution was well fitted by a single exponential distribution, which is the theoretical expectation for a population undergoing stochastic division as inferred from the distribution of division numbers above, and thereby confirmed that the manual ruleset was a valid approximation (S4H Fig). It is noteworthy that the average clone size of 4.5 cells (2.1× clone division rate) resulting from manual clone assignment (Fig 4A) did not match the expected average of 16-cell clones (4× clone division rate) based on the total liver cell numbers between the time of recombination and analysis (S2H Fig) [38], suggesting that likely larger clones were missed based on our strict, experimentally derived rule set. An overrepresentation of small clones caused by later than expected labelling events due to lingering PhiC31 protein represents a formal possibility, however, is less likely given that PhiC31 recombination occurs quickly and efficiently (S2E and S2F Fig). In summary, the unbiased mathematical approach confirmed that the manually defined clones and the resulting cell numbers are not or only minimally influenced by anisotropic tissue rearrangements in vivo.

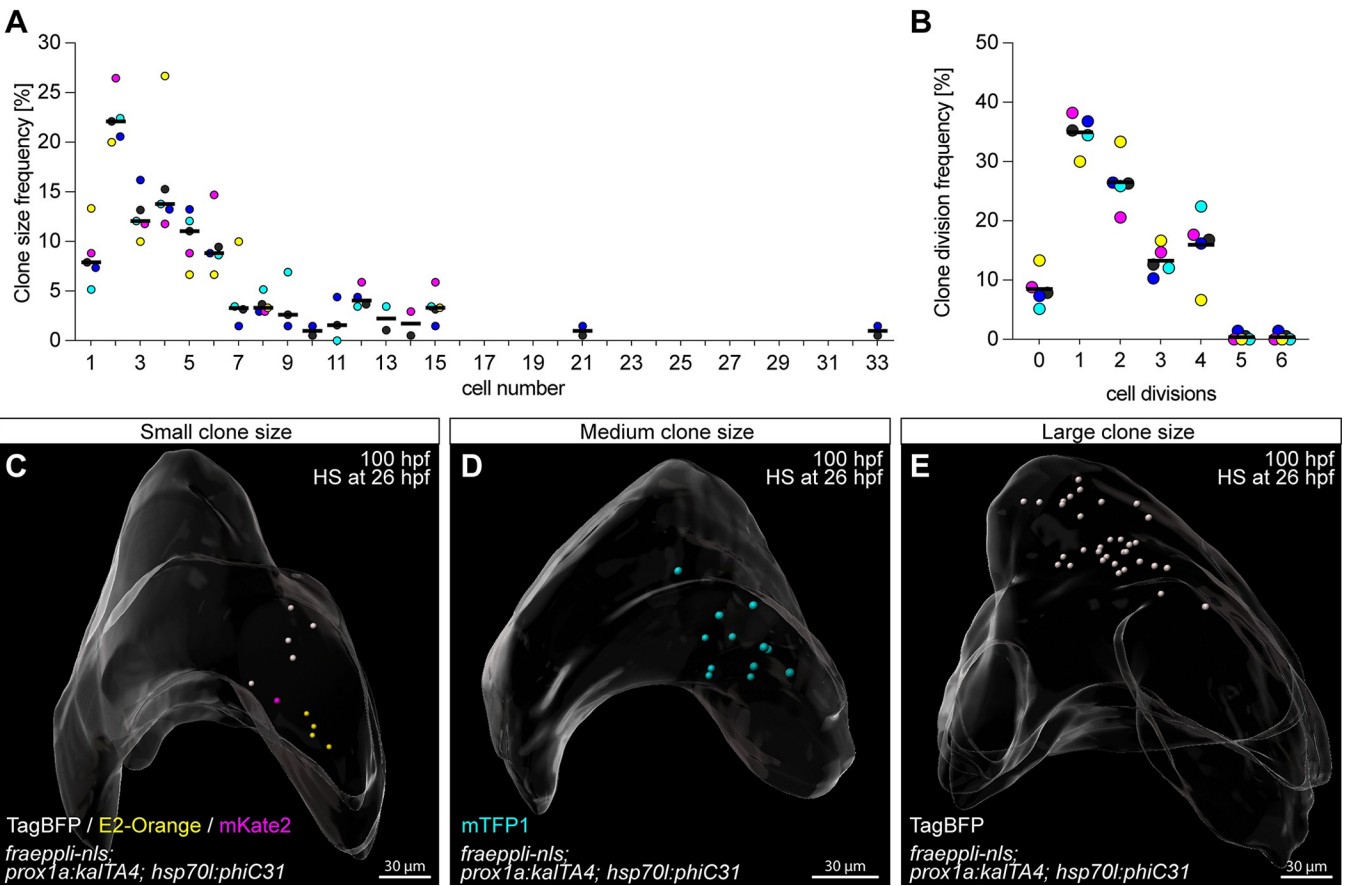

**Fig 4. Quantitative lineage tracing of hepatoblasts during embryonic development identifies heterogeneous growth behaviour.** (**A**) Frequency of manually assigned pure hepatocyte clone sizes ($N = 6$, $n = 190$ clones). (**B**) Distribution of the corresponding number of cell divisions for each pure hepatocyte clone ($N = 6$, $n = 190$ clones). (**A, B**) Clone colours are plotted in blue (TagBFP), turquoise (mTFP1), magenta (mKate2), and orange (E2-Orange); the mean of all colours is represented in black. (**C**) Whole-mount of a 100-hpf liver showing several clones, including a mKate2$^+$ 1-cell clone ($N = 6$, $n = 15$ livers). (**D**) Liver with a medium size 12-cell mTFP1$^+$ clone ($N = 6$, $n = 7$ livers). (**E**) Whole-mount of a 100-hpf liver with a large 33-cell TagBFP$^+$ clone ($N = 1$, $n = 1$ livers). (**C-E**) Labelled cells are represented as segmented nuclei, and an overall segmentation of the whole liver tissue is shown in transparent grey. The numerical values that were used to generate the graphs in (**A, B**) can be found in S1 Data.

## Embryonic cells contribute heterogeneously to postembryonic growth

Next, we asked how constituent progenitors contribute to postembryonic organ growth as the liver dramatically increases in size and changes shape, including the de novo formation of a third liver lobe (Fig 5A). Long-term lineage tracing experiments were performed to investigate whether the growth contribution of individual hepatoblasts is uniform throughout the organ, or whether some contribute minimally while others greatly. Similar to the lineage tracing experiments during development, we employed the FRaeppli-NLS system in combination with *hsp70l:phiC31; prox1a:kalTA4* to induce labelling of hepatoblasts by heat shock at 26 hpf (Fig 5B), followed by qualitative analysis of the spatial patterns clusters exhibit in postembryonic livers. For that, we acquired 3D datasets of 79 adult *fraeppli-nls; prox1a:kalTA4; hsp70l: phiC31* livers. In these recombined livers, a group of labelled cells is termed "cluster," since we cannot exclude that cells labelled in the same colour are the progeny of more than one hepatoblast. First, 11.4% of recombined livers displayed clusters that distribute along the central veins in the core of the liver lobe (Fig 5C–5C'). In most of these cases, clusters are oriented along the anterior–posterior axis of the fish. Second, in 30% of recombined livers, clusters distribute in a

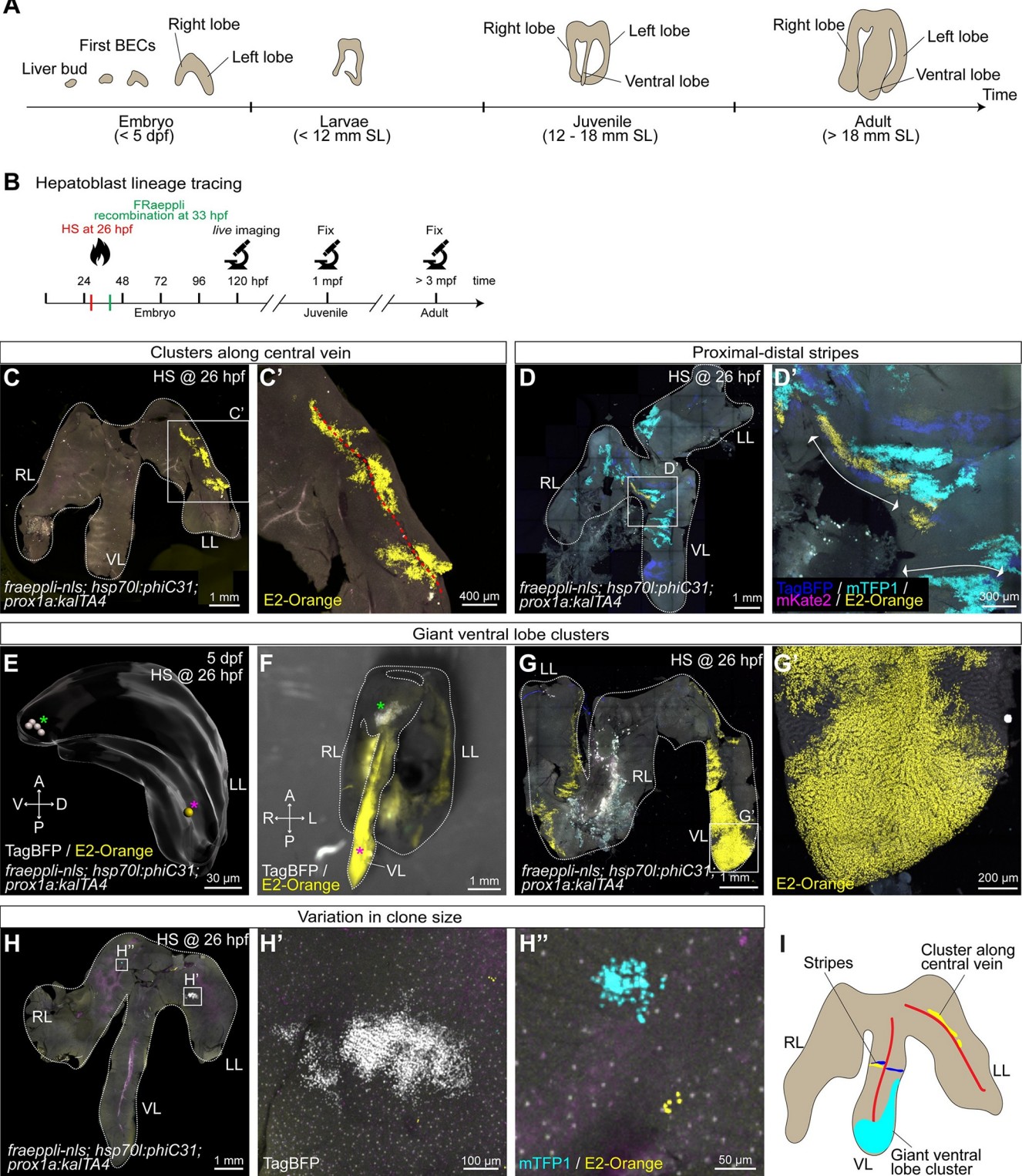

**Fig 5. Lineage tracing reveals heterogeneous cluster topologies during postembryonic growth.** (**A**) Schematic depicting key stages in postembryonic zebrafish liver development. (**B**) Experimental schematics of long-term lineage tracing experiments using *fraeppli-nls* embryos, inducing recombination by heat shock at 26 hpf to label hepatoblasts. At 120 hpf, embryos were screened by live imaging at the confocal microscope, and only sparsely labelled embryos were raised and fixed in either juvenile or adult stages. (**C-H**) Recombined livers showed different cluster topologies: clusters along central veins (**C-C'**) (*n* = 9 livers), proximal–distal stripes (**D**) (*n* = 23 livers) or giant clusters in the ventral lobe in adult (**F-G'**) (*n* = 3 livers). Large clusters in the ventral lobe can

originate from one single-labelled cell at 5 dpf (*n* = 1 liver) (**E**). (**F**) Stereomicroscope image showing the spatial location of the giant clone originating from a single recombined cell (**H**). Recombined livers show a range of cluster sizes from small (**H'**) to medium (**H''**). (**I**) Schematics of characteristic cluster topologies in recombined livers. Red lines indicate the blood vessel orientation in the liver. (**C-H**) Total numbers are (*N* = 9, *n* = 79 livers). A, anterior; P, posterior; R, right; L, left; RL, right lobe; LL, left lobe; VL, ventral lobe.

stripe-like fashion, consistent with cells proliferating and arranging along endothelial sinusoids (S5G Fig), creating parallel interspaced stripes. Specifically, in the lobe core, proximal–distal stripe-like clusters are oriented perpendicular to the central vein, such that labelled cells extend from the central vein to the margin of the lobe (Fig 5D–5D'). Similarly, striped clusters were also present in the anterior part of the liver, which connects the 3 lobes (S5A and S5A' Fig). Finally, 3.8% of recombined samples exhibited some unexpectedly large clusters, which we termed "giant clusters," since they occupied nearly half a lobe (Figs 5F–5G', S5B and S5C). Remarkably, giant clusters comprised 6% to 11% of the total liver volume, in contrast to an expected 0.7% assuming that all hepatoblasts at the time point of labelling proliferate equally. In contrast to the other 2 cluster shapes, which can be found throughout all 3 lobes, giant clusters were located exclusively at the tip of the ventral lobe and extended towards the left lobe (Fig 5F and 5G). For one giant cluster, we traced the embryonic origin to a single mOrange2-labelled cell at the periphery of the left lobe at 5 dpf (Fig 5E and 5F). Importantly, a TagBFP clone in the same sample forms a much smaller clone in the adult liver in the anterior ventral lobe (Fig 5E and 5F), suggesting that the variability in clone size is related to its position. Cluster size was generally heterogeneous across samples (Fig 5H–5H''), indicating that not all hepatoblasts contribute equally to the adult organ (Fig 5I).

To distinguish whether cluster shape and size are inherent to progenitors or influenced by other external factors, we performed a second set of experiments in which recombination was induced in differentiated hepatocytes. For this, *fraeppli-nls; fabp10a:kalTA4* were crossed to *hsp70l:phiC31* and administered a heat shock at 4 dpf for sparse labelling, and 42 juvenile and 31 adult livers were analysed. In 16% of adult livers, clusters extended along the large central vein (S5D and S5D' Fig) and 32% exhibited proximodistal stripes (S5E and S5E' Fig), consistent with the results of recombined hepatoblasts. At the same time, we could not detect any giant cluster in the ventral lobe. However, within the juvenile samples induced at 4 dpf, we identified in a single instance, representing 2.3%, a giant cluster in the ventral lobe, comprising 1.7% of the total liver volume (S5F and S5F' Fig). Hence, suggesting that hepatocytes can also contribute giant clusters to the ventral lobe. Due to the lower sample number, compared to the hepatoblast tracing, we did not observe this topological class in adults induced at 4 dpf. Interestingly, we did not observe any proximodistally striped clusters in juvenile livers, suggesting that growth along this axis predominantly occurs in late juvenile stages.

To assess the specificity of *hsp70l:phiC31* expression in noninduced embryos, we examined 100 juvenile or adult control livers and concluded that nonspecific recombination is rare (S5H–S5J Fig). Overall, these findings suggest that cluster shapes and growth patterns are independent of the intrinsic proliferation potential of single hepatoblasts, or the differentiation state of the recombined cell, and instead are the result of extrinsic growth signals.

To investigate hepatic growth dynamics, we examined whether proliferating cells are evenly distributed or spatially enriched during development. Interestingly, from 84 hpf onwards, proliferating cells are located significantly closer to the surface than the centre of the liver (S6A and S6B Fig), suggesting that growth is enhanced at the organ periphery. Measuring the distance to the nearest neighbours between proliferating hepatocytes and BECs or across each cell population revealed that average distances between proliferating cells are greater than between all cells of the given population (S6C and S6D Fig), rejecting the possibility of proliferation clusters.

## Polyploid cells appear transiently during postembryonic liver growth

Postembryonic growth (S7A Fig) is poorly understood across organs and species. Monitoring liver growth by comparing liver to body weight (S7B and S7C Fig) between late larval to adult stages revealed a two-phase process: The liver grows faster than the body in juveniles until it attains a stable liver to body weight ratio once the fish reach adulthood, with an average of 5.5% for female and 3.5% for male zebrafish (S7D Fig). However, the proportion of liver to body weight in juveniles is higher, on average 8.6%, and more variable (2.1% to 22.2%; S7D Fig), likely reflecting substantial organ growth including the formation of an additional lobe.

From a mechanistic point of view, rapid growth could be achieved by increasing cell size, cell ploidy, and proliferation, including de novo tissue extensions. First addressing whether polyploid hepatocytes are present in the zebrafish liver, particularly during periods of rapid organ growth, we analysed whole-mount livers of *fabp10a:GFP; tp1:H2B-mCherry* counterstained with 4′,6-diamidino-2-phenylindole (DAPI). At the larval stage, at 14 dpf, shortly after liver function commences, the volume of hepatocyte nuclei was variable, and rare binucleated hepatocytes appeared in 50% of embryos (Fig 6A–6A"). Subsequently, in juveniles, we observed 2 scenarios: multinucleated hepatocytes and enlarged nuclei (Fig 6B and 6B'), with an up to 10-fold volume increase and elevated DAPI intensity compared to average hepatocyte nuclear values (Fig 6C and 6D), indicating that zebrafish hepatocytes enter a polyploid state. Contrary to larval livers, all juvenile samples exhibit enlarged nuclei and multinucleated cells at high frequency in all 3 lobes. Next, we assessed whether polyploid hepatocytes are maintained, become more abundant, or disappear with age. Surprisingly, we detected only a single group of 3 polyploid nuclei in one of 5 adult livers (Fig 6E–6G), showing that hepatocyte polyploidy is transient, as it peaks with the massive growth phase of the zebrafish liver and declines with maturation (Fig 6H).

## Liver morphology and clonal growth patterns reveal distinct ventral lobe formation

Assessing whether other mechanisms besides ploidy may contribute to postembryonic liver growth, such as extension by additional organ parts, we turned our attention to the de novo–forming ventral lobe, because of the unique giant clusters and their distinct clonal growth behaviour. The ventral lobe is described as arising from the ventral part of the liver [15], yet experimental data concerning its origin and formation are missing. Therefore, to relate the origin of the giant clusters to ventral lobe formation, we carefully examined liver morphology throughout postembryonic growth in the 62 larval and juvenile livers collected for the lineage tracing studies. Based on morphological characteristics, ventral lobe formation across postembryonic growth was divided in 6 phases (Fig 7A). During stage I, the liver consists mainly of the right and left lobe, with the latter exhibiting a slight bulge towards the ventral midline (Fig 7B). During stage II, the ventral lobe has started to form and identifies with a very thin structure that originates in the more posterior half of the left lobe (Fig 7C). The position of ventral lobe outgrowth shifts to the more anterior part of the left lobe during stage III, while it still maintains its long thin and flat appearance (Fig 7D). The tip of the ventral lobe starts to round up and expands in a radial manner during stage IV, and at the same time, the base of the ventral lobe broadens, strengthening its connection at the ventral most part of the liver (Fig 7E). Thereafter, during stage V, the ventral lobe increases in size expanding laterally (Fig 7F), reaching its final width in late juvenile stages or early adulthood (stage VI, Fig 7G).

Using these morphological data as a reference, we turned to the long-term lineage tracing data asking whether the future ventral lobe would arise from the embryonic left lobe. We correlated characteristic ventral lobe cluster patterns in juvenile and adult livers, with

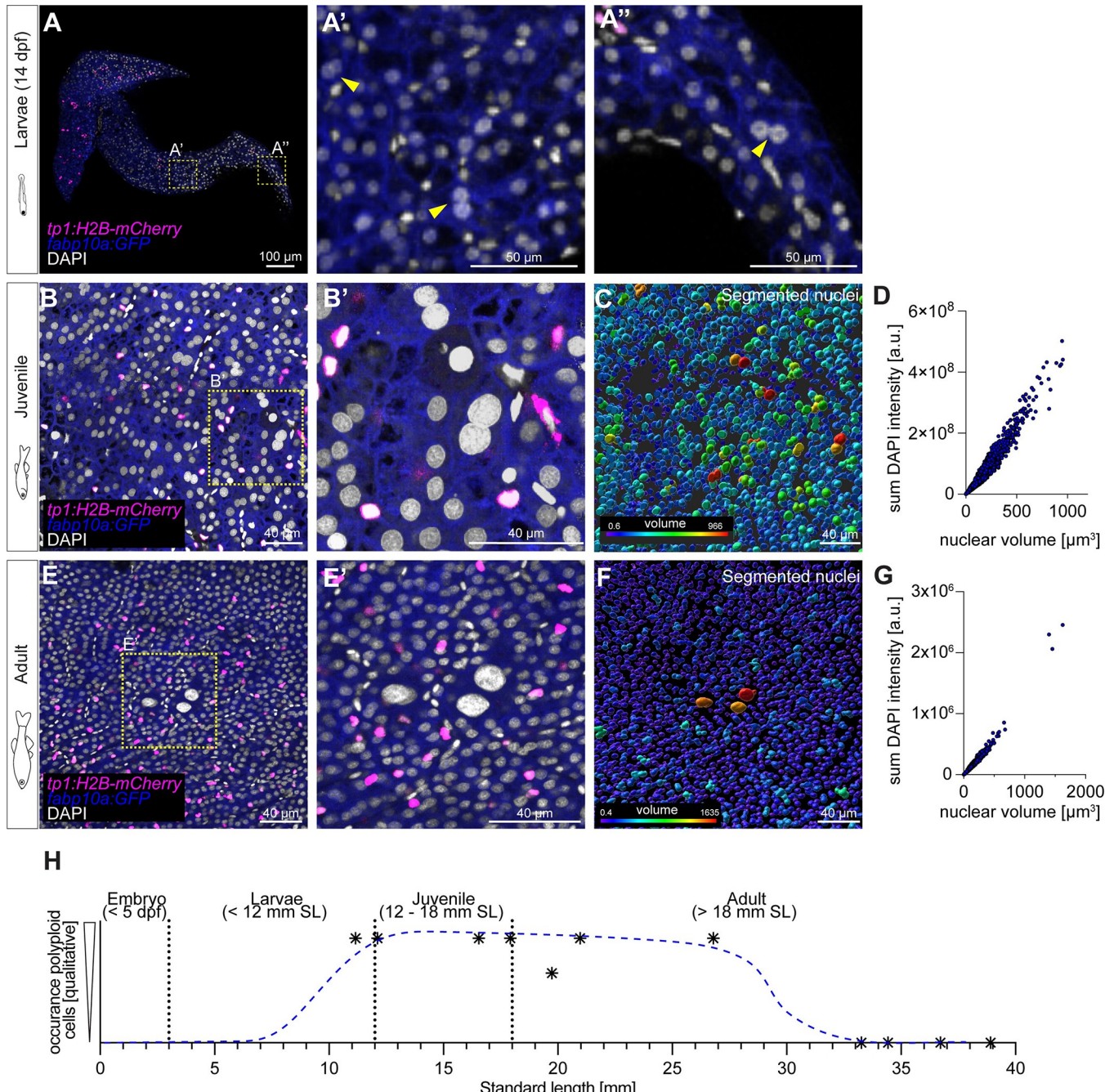

**Fig 6. Polyploid cells appear transiently in hepatic postembryonic growth in zebrafish.** (**A**) Whole-mount of a 14-dpf zebrafish liver, displaying sparse multinucleated hepatocytes (*N* = 1, *n* = 2 livers; yellow arrowheads indicate binucleated cells). (**B**) Approximately 5 μm projection of a region of a juvenile liver. Fish SL = 11.16 mm (*N* = 3, *n* = 6 livers). (**C**) Segmentation shows variable nuclear volumes, which correlate with the sum intensity of DAPI, indicating that bigger nuclei have a higher amount of DNA (**D**). (**E**) Approximately 5 μm projection of an adult liver region (*N* = 3, *n* = 3 livers). (**F**) Segmented nuclei show only sparse variability in volume, with few bigger nuclei. Nuclear volume correlates with sum intensity of DAPI (**G**). (**H**) Schematics representing the transient appearance of polyploid cells over time; blue trajectory is manually approximated based on qualitative analysis. The numerical values that were used to generate the graphs in (**D**, **G**, **H**) can be found in S1 Data. DAPI, 4′,6-diamidino-2-phenylindole; dpf, day postfertilization; SL, standard length.

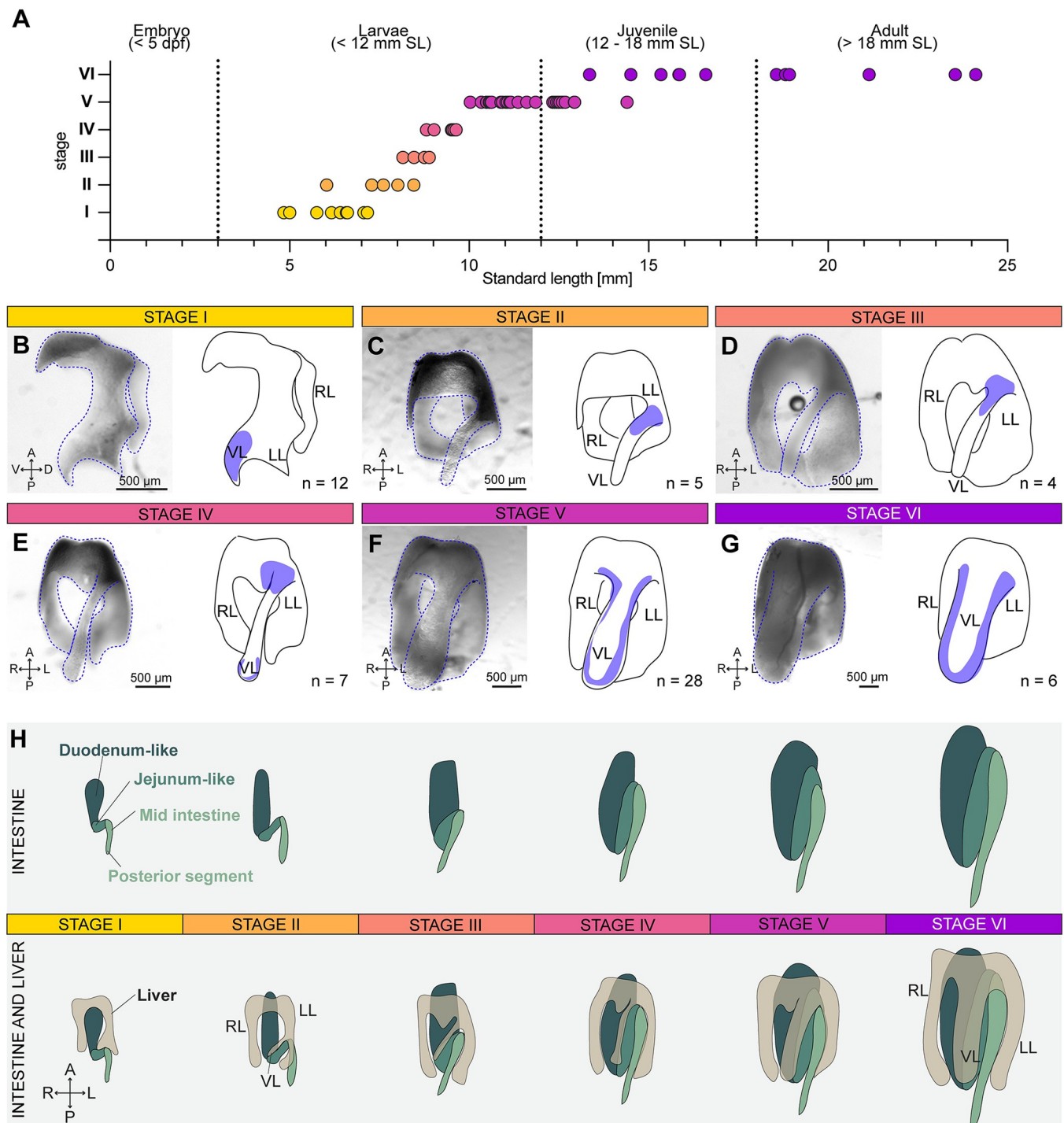

**Fig 7. Ventral liver lobe formation during postembryonic growth.** (A) The 6 steps of ventral liver lobe formation correlate with fish standard length (SL). The numerical values that were used to generate the graph can be found in S1 Data. (**B**) Stage I: A small tissue extension at the tip of the left lobe is visible (*n* = 12 livers). (**C**) Stage II: a thin ventral lobe originates in the lower half of the left lobe (*n* = 6 livers). (**D**) Stage III: the thin ventral lobe shifts position towards the more anterior part of the left lobe (*n* = 4 livers). (**E**) Stage IV: the tip of the ventral lobe starts to expand (*n* = 7 livers). (**F**) Stage V: lateral-oriented expansion of the ventral lobe (*n* = 28 livers). (**G**) Stage VI: enlargement of all lobes in width (*n* = 6 livers). The blue areas in the schematics mark the region characteristic for the respective stage. (**H**) Schematic depicting the morphology of the liver in relation to the folding of the intestine in stages I-VI. A, anterior; P, posterior; R, right; L, left; RL, right lobe; LL, left lobe; VL, ventral lobe.

corresponding clonal positions within the same fish acquired at 5 dpf. For example, a 3-cell mKate2$^+$ clone located at the periphery of the left lobe at 5 dpf (S8A Fig) grew into a 1,420-cell mKate2$^+$ clone at the juvenile stage, exclusively localised in the ventral lobe (S8B Fig). Furthermore, 36% of juvenile livers contained clusters displaying a similar orientation from the left lobe into the tip of the ventral lobe across stages (S8C and S8D Fig). These distinct and stereotypic clone patterns strongly support our hypothesis that the ventral lobe grows out from the left lobe.

Interestingly, the most dramatic morphological changes occur during the larval to juvenile transition (stages I to IV), mirroring zebrafish metamorphosis, during which many organs transform and adopt adult characteristics, such as the gut [17]. Considering the overall morphological changes occurring during zebrafish metamorphosis, the morphogenesis of the intestine, in particular the appearance of the 2 intestinal bends, coincides temporally and spatially with the repositioning of the ventral lobe (Figs 7H and S8E–S8P). The intestine and liver are in direct contact during those stages, with the ventral lobe situated directly on top of the first intestinal fold. Notably, when the gut starts folding, ventral lobe formation is initiated (stage I) (S8E and S8F Fig). With progressive bending of the gut, the position of the outgrowing ventral lobe shifts anteriorly (stages II to V) (S8G–S8N Fig). In parallel, as the ventral lobe expands laterally (stage VI), it is located directly above the intestinal fold (S8O and S8P Fig). Finally, in adults, the 3 liver lobes almost entirely enwrap the intestine. These data suggest that gut and liver morphogenesis are coupled during metamorphosis (Fig 7H), in line with the idea of stimulating cues from the intestine, promoting the selective expansion of the ventral lobe, as mirrored by the spatial expansion dynamics of the giant clusters.

## Discussion

The contribution of multipotent progenitor cells to organ formation and the establishment of correct cell proportions is vital for building a functional organ. This study shows how heterogeneous hepatoblasts contribute to fate decisions and postembryonic liver growth (Fig 8A and 8B), including de novo structures, enabling the formation of a functional liver architecture. We show that a precise 1:9 BEC-to-hepatocyte ratio within the liver is established and remains nearly constant into adulthood. Cell counts from single-cell transcriptome studies of adult zebrafish match this ratio [39], corroborating that specific and constant cell proportions are pivotal for a functional architecture across all stages. Notably, this distinct ratio is already established before hepatic cells take up their function, allowing time for the morphogenetic differentiation of the intricate liver architecture, reminiscent of stem cell–based tissue homeostasis in the intestine where an early lineage commitment leading to a precise cell type ratio is thought to facilitate maturation prior to function [40].

The establishment of a precise cell type ratio may be controlled by spatial signalling, such as the Notch pathway [41], given the seemingly even spacing of the first appearing BECs (Fig 2F). Both in zebrafish and mice, Notch signalling plays a central role in BEC fate [28,33,42]. Similar to zebrafish, hepatocytes vastly outnumber BECs not only in homeostatic human livers but also in mice and rats [9,43]. With approximately 1:32 BECs to hepatocytes [9,44], these magnified proportions in the mouse may be explained by its tissue organisation in which not every hepatocyte directly connects to bile ductules like in zebrafish, so fewer BECs may be required for the overall network.

We combined in silico modelling with in vivo lineage tracing experiments to clearly show that lineage contributions of progenitor cells, and not differences in their proliferation rates, determine the given cell type ratio. Notably, transiently changed BEC proliferation between 96 and 120 hpf, which would push the cell type ratio towards BECs, did not noticeably alter

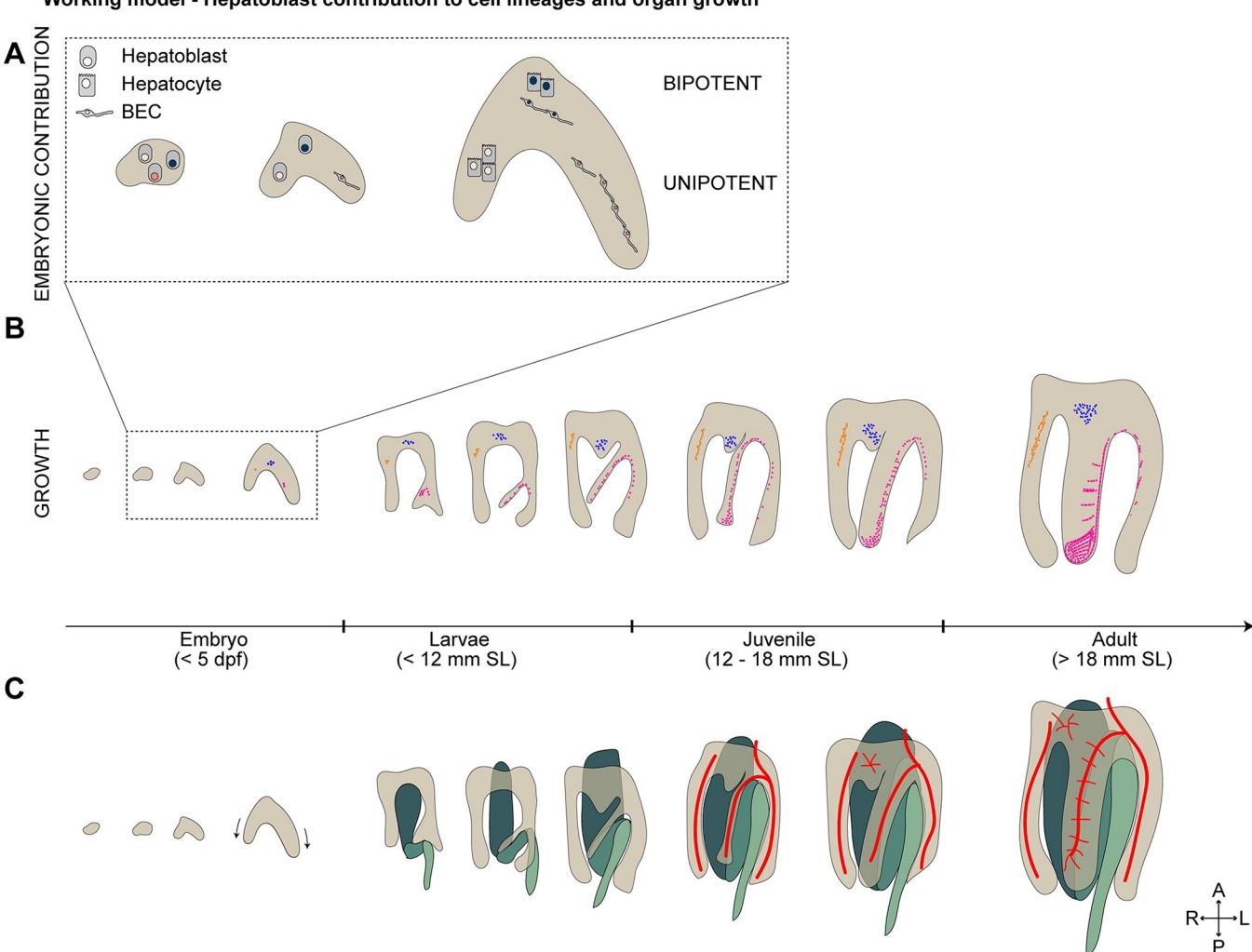

**Fig 8. Working model of hepatoblast contribution to lineage decision and postembryonic growth.** Schematics showing the current working models: (**A**) uni- and bipotent hepatoblast contributions to hepatocytes and BECs following heterogeneous lineage decisions. (**B**) Hepatoblasts contribute with heterogeneous proliferation behaviours to postembryonic liver growth. Cells from the embryonic left lobe contribute to the ventral lobe, including the formation of giant clusters (magenta). (**C**) The liver morphology changes dramatically simultaneous to the intestinal bending occurring during postembryonic growth (green). BEC, biliary epithelial cell; dpf, day postfertilization; SL, standard length.

proportions, suggesting a mechanism that controls variances in cell type proportions to maintain a stable tissue architecture and function. We did not model cell death and associated cell competition as contributing factor since it has been previously shown that there are no or rare apoptotic cells in the liver during early development [38,45,46]. These results unequivocally demonstrate the presence of bi- and unipotent lineage decisions in vivo (Fig 8A). Based on in vitro studies and lineage tracing of early foregut endoderm cells induced at E7.75, it has long been assumed that all hepatoblasts are bipotent and contribute to both cell lineages [8,10]. However, murine lineage tracing experiments of Lgr5+ hepatoblasts revealed functional heterogeneity of hepatoblasts [9], contributing equally to pure hepatocyte or mixed hepatocyte and BEC clones adjacent to the portal triads [9]. Our analysis of clones throughout the liver using the pan-hepatoblast driver *prox1a:kalTA4* revealed a larger proportion, 88.8%, of unipotent hepatocyte clones. Remarkably, we also uncovered pure BEC clones indicating a rare

population of unipotent BEC differentiation. Our result thereby provides lineage-tracing evidence for the presence of early BEC-committed hepatoblasts, which have so far only been proposed based on single-cell RNA sequencing of murine hepatoblasts at E10.5 [9]. Our study shows uni- and bipotent hepatoblasts in zebrafish, demonstrating that heterogeneous lineage potential at the start of liver formation is conserved across species. We propose progenitor heterogeneity as a general strategy to set up lineage proportions in the liver, yet the underlying molecular mechanisms defining instructive signalling hierarchies [9] may differ, since an Lgr5 homologue seems to be missing in most teleost genomes, including zebrafish [47].

Growth dynamics of embryonic and especially postembryonic liver growth are poorly understood across species, including zebrafish. Here, we show a steady decrease in BEC and hepatocyte proliferation, in line with the notion from other organs that high proliferation in undifferentiated tissues opposes lower proliferation rates in differentiated tissues [48]. Proliferating cells were enriched at the liver periphery, suggesting that external signals fuel tissue growth at the edge once differentiation ends and tissue architecture is established. Peripheral liver growth is regulated by β-catenin signalling in chicken [25], and Wilms tumour 1 from surrounding mesothelial cells promotes murine liver growth and lobe formation [23,24]. We show that the liver grows exponentially during metamorphosis and provide, to our knowledge, the first evidence for polypoid hepatocytes in the zebrafish, well described in most mammals [49]. Both large nuclei and multinucleated cells are prominent in juvenile livers, yet rare in adults, which is markedly more similar to the homeostatic 20% to 40% polyploidy in humans compared to the 75% to 94% in mice [49,50]. Polyploidy represents an attractive developmental strategy to quickly increase organ size while maintaining or even elevating metabolic function due to higher DNA content and enlarged cell volume [51,52]. Nevertheless, the overall link between polyploidy and growth remains unclear, since inhibiting the formation of polyploid hepatocytes does not impair liver growth [53] and their relevance for liver regeneration remains controversial [54–56].

We provide the first qualitative study of postembryonic liver growth by tracing hepatoblast- and hepatocyte-progeny thereby identifying distinct growth patterns (Fig 8B). Remarkably, 2 cluster categories arranged along blood vessels, one paralleling the central vein and the other the perpendicularly organised sinusoids. This is in line with vessel-derived mitogens that may control hepatocyte proliferation, such as Wnt2, Angiopoetin, and Hepatocyte growth factor, as well as signalling induced by mechanical stretching of endothelial cells by the incoming blood [57–60]. These cellular relationships further agree with the finding that hepatocytes and their newly arising daughter cells align parallel to sinusoids after hepatocyte damage [61] as well as cohesive and oriented growth at the organ surface during lobe development [62]. Lastly, the finding of giant clusters associated with ventral lobe growth is surprising, as it suggests growth dynamics and signals distinct from the 2 dorsal lobes. Strikingly, the correlation between adult clonal growth patterns and their embryonic origin demonstrates that the ventral lobe arises surprisingly from the embryonic left lobe and not in the most ventral part, as previously suggested [15]. Given all 3 lobes are of similar size in the adult liver, the overproportionate contribution of these giant clusters to the mature ventral lobe, therefore, suggests its origin from far fewer embryonic cells than the 2 dorsal lobes, which is reminiscent of the clonal dominance observed in other tissues [21,63,64]. The stereotypic pattern of the giant clusters within the ventral lobe further indicates localised growth driving postembryonic organ morphogenesis. Signals could arise from the spatially and temporally correlating remodelling of the intestine and markedly the formation of its 2 bends (Fig 8C). The intestine may provide physical constraints, metabolites, and/or mitogens directing the characteristic growth of the ventral liver lobe. Interorgan communication, such as for the *Drosophila* testes and intestine [65], represents an attractive strategy for guiding the distinct formation of the ventral liver lobe.

Understanding differential modes of lobe formation is also highly relevant for regeneration studies in adults, since due to its accessibility, the ventral lobe is targeted for partial hepatectomy in zebrafish [66,67]. Yet, depending on the extent of injury, the liver responds with either epimorphic or compensatory regeneration [26]. The underlying mechanisms remain elusive; thus, further studies of postembryonic liver growth and, in particular, ventral lobe formation are pivotal.

In summary, we show that the heterogeneous lineage contribution of hepatoblasts is the predominant factor establishing the distinct cell proportions of the functional liver, while heterogeneous proliferation dynamics of individual progenitors establish organ size. We propose that both lineage and proliferation heterogeneity is not an intrinsic hepatoblast property, but stochastic and influenced by signals from the microenvironment. Identifying the molecular nature of these signals, as well as the morphogenetic principles directing tissue architecture, will aid in developing strategies promoting the endogenous capacity of the liver to restore a functional tissue organisation and instruct engineering of hepatic tissues in vitro.

## Materials and methods

### Ethics statement

All experiments were performed according to ethical guidelines approved by the Danish Animal Experiments Inspectorate (Dyreforsøgstilsynet) with the approval number 2018-15-0201-01431.

### Zebrafish husbandry

Zebrafish *(Danio rerio)* embryos and adults were kept according to standard laboratory conditions [68].

The following transgenic strains were used: *tgBAC(prox1a:kalTA4)$^{uq3bh}$* [69], *tg(kdrl:EGFP)$^{s843}$* [70], *tg(EPV.TP1-Mmu.Hbb:hist2h2l-mCherry)$^{s939}$* [71], *tg(-2.8fabp10a:EGFP)$^{as3}$* [72], *tg(tp1-MmHbb:EGFP)$^{um14}$* [73], *tg(fraeppli-nls)$^{cph1-3, cph9}$* [32],*tg(fabp10a:kalTA4; cryaa:Venus)$^{cph8}$* [32],*tg(hsp70l:phiC31-integrase, he1a:lyn-Citrine)$^{cph7}$* [32], *tg(5xUAS:EGFP)$^{zf82}$* [74].

### FRaeppli activation

In all experiments unless otherwise stated, FRaeppli recombination was induced conditionally using *tg(hsp70l:phiC31-integrase, he1a:lyn-Citrine)$^{cph7}$* [32]. To induce *phiC31* expression, embryos were subjected to a 30-minute heat shock at 39˚C. Control embryos were kept at 28˚C. To avoid undesired recombination due to temperature fluctuations, embryos were stored at strict temperature control, including double Styrofoam boxes for transport of recombined embryos. Moreover, FRaeppli recombination was induced by *phiC31 integrase* mRNA (30 to 40 pg) injection into one-cell stage embryos (S2H and S2I Fig). To express FRaeppli colours in hepatoblasts and, subsequently, hepatocytes and BECs or only hepatocytes expression was controlled by either *tgBAC(prox1a:kalTA4)$^{uq3bh}$* or *tg(fabp10a:kalTA4; cryaa:Venus)$^{cph8}$*, respectively. We mainly used *fraeppli-nls$^{cph09}$*, which recombines more sparsely than *fraeppli-nls$^{cph01-03}$*. Unless stated otherwise, experiments were performed with transgenic *fraeppli-nls* lines carrying one insertion.

### Imaging

All embryos were raised in E3 medium (5 mM NaCl, 0.17 mM KCl, 0.33 mM CaCl2, and 0.33 mM MgSO4). Medium was supplemented with 0.2 mM 1-phenyl 2-thiourea (PTU, Sigma-Aldrich) for embryos intended for live imaging. During live imaging, embryos and larvae were

immobilised in 0.8% low melting agarose and anaesthetized with Tricaine (164 mg/L; MS-222, Sigma-Aldrich) dissolved in E3/PTU.

To identify sparsely recombined *fraeppli-nls* embryos at 120 hpf, 10 to 40 embryos were mounted and screened at the confocal microscope for the presence of recombined cells, using spectral imaging. Imaging time was kept as short as possible, and embryos were released from agarose immediately after imaging.

Vibratome sections of 300 μm vibratome sections or whole juvenile and adult livers were cleared with SeeDB2G [75]. Livers of zebrafish larvae with a SL smaller than 8 mm were directly mounted in Omnipaque350 (Sigma-Aldrich, Histodenz) for imaging. Fixed *fraeppli-nls* embryos were embedded in VectaShield (VWR, VECTH-100) for imaging. Imaging was performed using LSM 780 and 880 confocal microscopes equipped with PMT detectors for sequential imaging and a spectral detector (GaAsP-PMT, 32 channels, 410 to 694 nm range, 8.9 nm bandwidth) for spectral acquisition. Five-colour sequential imaging of adult recombined livers was performed with a Leica Stellaris confocal microscope equipped with a tuneable white-light laser, 488 and 448 nm diodes, and 4 PMT/HyD detectors. Adult recombined FRaeppli livers were imaged to 500 μm in depth using the 10× objective. Detailed parameters for sequential and spectral imaging of FRaeppli-NLS modes are described in Caviglia and colleagues [32]. Stained embryos embedded in benzyl alcohol–benzyl benzoate (BABB) (ratio 1:2) were imaged with a Leica SP8 confocal microscope.

Brightfield and fluorescent overview images of recombined *fraeppli-nls* livers were acquired at a Leica Stereomicroscope (Leica, M205 FCA) equipped with a CCD microscope camera (Leica, DFC7000 GT).

## Immunostaining

Embryos and larvae were fixed with 4% PFA at 4˚C overnight. For adult liver staining, the liver was embedded in 4% agarose and cut into 200 μm sections using a vibratome. Tissue sections were stained in 9-well glass plates. Immunostainings were performed as previously described [76]. In short, for nuclear staining, embryos were permeabilized using DNase I (Thermo Fisher Scientific) treatment for 45 minutes at 37˚C and then blocked in 10% donkey serum (Jackson Immunoresearch, 017-000-121) with 1% PBS-Triton X-100. Primary antibodies were incubated with 1% PBS-Triton X-100, 1% dimethylsulfoxide (DMSO) and 10% donkey serum at 4˚C for 1 or 2 nights slowly shaking.

Primary antibodies included α-Hnf4α (goat polyclonal antibody, 1:100, Santa Cruz, sc-6556); α-mCherry (rat monoclonal antibody, 1:1,000, Invitrogen, m11217); α-Prox1 (rabbit polyclonal antibody, 1:500, Angiobio, 11-002P); and α-2F11 (mouse monoclonal antibody, 1:1,000, gift from Julian Lewis). After washing in 0.1% PBS-Triton X-100, secondary antibodies were incubated with 1% PBS-Triton X-100, 1% DMSO, and 10% donkey serum at 4˚C for 1 to 2 nights.

Secondary antibodies included donkey-anti-goat 488 (1:200, Jackson Immunoresearch, 705-545-147), donkey-anti-rat-cy3 (1:500, Jackson Immunoresearch, 715-166-150), goat-anti-rabbit 647 (1:500, Jackson Immunoresearch), and donkey-anti-mouse 647 (1:500, Jackson Immunoresearch, 715-605-151). For DNA staining, samples were incubated with DAPI (1:100 to 500, Sigma, D9542) or TO-PRO-3 (1:50,000, Thermo Fisher Scientific) for 1 hour at room temperature or overnight at 4˚C. Samples were washed in 0.1% PBST and dehydrated in methanol.

## Processing juvenile and adult zebrafish

Juvenile and adult zebrafish were fixed as whole fish with 4% PFA at 4˚C for more than 20 hours using a rotator. To allow PFA penetration into deep tissues, the skin of the abdominal

wall was opened prior to fixation. After fixation, samples were washed in PBS for at least 1 hour at room temperature using a rotator. An overview picture was taken of each fixed fish to determine the SL [16] by measuring the distance from the mouth to the start of the fin with Fiji. Whole fish weight was assessed using a precision balance (Sartorius, Qunintx). The liver was dissected and weighed again in a preweighed Eppendorf tube filled with 1 ml of PBS to avoid errors in the low weight range of the scale. For confocal imaging, livers were mounted flat between 2 coverslips with clay spacers or on a slide with 2 imaging spacers. We observed a large variation in liver shape with respect to lobe length and size, although the ventral lobe is consistently flatter than the 2 dorsal ones. In the majority of fish, the left and right lobes are merged dorsally at the posterior tips of the lobes and show connected vascular networks, whereas others display 3 independent lobes only connected at the most anterior part, suggesting that liver growth and organ morphogenesis is plastic. When the tissue was connected between the tips of the left and right lobes, the connecting tissue was carefully cut with forceps to ensure flat mounting.

### Image analysis and cell segmentation

Images were processed with Bitplane Imaris and Zeiss ZEN software. As default, maximum intensity projections are displayed, unless indicated differently in the figure legends. Lambda scans were unmixed using HyperSpectral Phasor software (HySP) (version 0.9.10) [77]. A workflow describing the unmixing of FRaeppli FPs with HySP has been published in detail [32].

For all embryonic images, the automated spot detection function in Imaris was used for quantification of cell numbers. EdU-positive cells of each cell type have been identified by segmenting EdU-positive nuclei and filtering for expression level of either Hnf4 (hepatocyte) or *tp1*:*mCherry* (BEC). The liver volume was segmented by manual surface generation in Imaris based on morphological difference of the liver tissues compared to surrounding tissues. Distances of cells to the liver surface was extracted in Imaris by using Spot-to-Surface distance. Distances of cells to nearest neighbours were also automatically obtained after segmentation in the latest version of Imaris, Imaris 9.9. To determine ploidy, nuclear volume was segmented using surface function in Imaris and Sum Intensity of DAPI measured within the segmentation.

To determine the ratio of endodermal cell types in adult tissues, we investigated 4 juvenile and 2 adult livers stained with DAPI. Of each juvenile liver and one adult, we extracted at least 3 regions of interest (ROIs) of whole-mount images acquired with a 40× objective with a size of $1,000 \times 1,000 \times 50$ μm and segmented the nuclei. In addition, one adult liver was sectioned in 300 μm thick agarose sections using the vibratome and at least 2 to 3 ROIs from each of the 3 sections were selected. When available, ROIs taken from different lobes were included. For BEC nuclei, we used the *tp1*:*H2B-mCherry* signal in the 564-nm channel for segmentation. For quantifying hepatocytes, DAPI signal was segmented and filtered against high-intensity cells to exclude blood cells and for expressing GFP from *fabp10a*:*GFP* transgene. Although we observed that the localization of the endogenous fabp10 signal varied between samples, from a homogeneous cytoplasmic signal to a more peripheral localization, the signal was sufficient to identify hepatocyte nuclei by determining the overlapping signal of DAPI with GFP.

For embryonic samples, *fraeppli-nls*-marked cells were segmented using the Imaris spot function. The xyz coordinates were extracted and subsequently used for the mathematical analysis of cell fate assignment in clones. BEC nuclei were identified by filtering spots for the expression of GFP (*tp1*:*EGFP*). In embryos without *tp1*:*EGFP*, the elongated nuclear shape was used as a proxy for assigning BEC fate. Only very elongated nuclei were categorised as BEC. In case of doubt, the clone was excluded from the analysis.

For adult recombined *fraeppli-nls* livers, images were binned in Fiji by factor 2 in x and y to reduce image size. Subsequently, clonal volume was quantified in Imaris using the surface function.

## EdU incorporation assay and proliferation analysis

To label proliferating cells, embryos were incubated with 400 μM EdU (Invitrogen)/DMSO in fish water for 1 hour at 28˚C. Dependently on the stage, different DMSO concentrations were used: 48 to 60 hpf: 15% DMSO; 72 hpf: 10% DMSO; 84 to 144 hpf: 5% DMSO. Upon incubation, embryos were immediately fixed in 4% PFA. EdU-labelled cells were detected with the Click-it EdU Imaging Kit Alexa Fluor 647 (Invitrogen). Proliferating cell numbers were assessed using the Imaris software.

## Mathematical model of embryonic liver development

For simulating liver development, 3 distinct phases were included: proliferating hepatoblasts (phase I), differentiation into hepatocytes and BECs (phase II), followed by hepatocyte and BEC proliferation (phase III).

The algorithm follows the steps listed below:

First, we start with an initial number of cells in the hepatoblast state ($n = 100$), with all cells proliferating at the same cell cycle length. Second, once hepatoblasts reach a specific number of cells ($n = 200$), differentiation into hepatocytes and BECs is initiated. Cell differentiation is always linked to cell division of hepatoblasts. For the 4 different models, distinct probabilities for hepatoblast differentiation into hepatocytes and/or BEC were defined.

|  | Model 1 | Model 2 | Model 3 | Model 4 | Model 5 | Model 6 |
| --- | --- | --- | --- | --- | --- | --- |
| p_divide [P ➜ H + C] | 1 | 1 | 0.5 | 0.2 | 0.2 | 0.2 |
| p_divide [P ➜ C + C] | 0 | 0 | 0 | 0.05 | 0.05 | 0 |
| p_divide [P ➜ H + H] | 0 | 0 | 0.5 | 0.75 | 0.75 | 0.8 |

Third, once cells acquire the hepatocyte or BEC fate, they continue to divide while maintaining the acquired fate. The cell type–specific division rates are defined for each model as follows, whereas 10 (a.u.) corresponds to 24 hours.

|  | Model 1 | Model 2 | Model 3 | Model 4 | Model 5 | Model 6 |
| --- | --- | --- | --- | --- | --- | --- |
| division rate_ P | 1/10 | 1/10 | 1/10 | 1/10 | 1/10 | 1/10 |
| division rate_ C | 1/10 | 1/15 | 1/10 | 1/10 | 1/15 | 1/10 |
| division rate_ H | 1/10 | 1/10 | 1/10 | 1/10 | 1/10 | 1/10 |

The simulation is terminated when 3,000 cells are reached.

For nonsynchronized, random cell divisions, we used a Gillespie algorithm to calculate the division time for each individual cell. The model is a 3D agent-based model where a cell divides by placing a new (daughter) cell close to itself. The positions of the cells are continuously being updated with the method previously developed by Nissen and colleagues [78].

The code was generated in Python. For each model, 10 simulations were performed, and a script in MATLAB R2022a was used for statistical analysis and generation of the plots.

## Manual quantification of clones during embryonic lineage labelling

**Determining clones.** We assigned a rule to manually assign recombined cells of the same colours to clones. Live imaging of *fraeppli-nls; hsp70l:phiC31; prox1a:kalTA4* embryos showed

that individual cells in the liver move a maximum of 20 μm apart and change position between 60 and 80 hpf (S2B Fig), at a time when hepatic fate is already established. Considering the total volume of the liver, which increases 3.2-fold between 72 and 100 hpf, and nuclear movements of 20 μm maximum, we estimated that adjacent cells could move 70 μm apart between 60 and 80 hpf. We manually assigned clones to include all recombined cells of one colour within a radius of 70 μm distance. The spot measurement function was used in Imaris to determine intercellular distance. Cells defined as clones were grouped together. The position (xyz coordinates) of each individual labelled cell was then subsequently extracted from Imaris for analysis of clonality. Due to difficulties in precise clone definition, 12.4% of livers with very dense labelling at 100 hpf were excluded from the analysis.

**Determining division frequency of clones.** Division rates were calculated based on clone size. Clones with cell numbers outside exact divisions were rounded down, e.g., a 5-cell clone was categorised as 2 divisions. Rounding up did not change the overall division distribution.

## Quantitative clonal reconstruction and assessment of clonality

For a single population of progenitor cells undergoing stochastic fate choice (e.g., division and apoptosis), 2 key theoretical predictions are expected to hold.

Firstly, given a constant division rate k, the probability distribution for a cell population to divide n times within a specific time frame T is predicted to follow a Poisson distribution: $P(n) = \frac{\lambda^n e^{-\lambda}}{n!}$, a distribution with $\lambda = kT$ being the single adjustable parameter, equal to the average number of divisions in this time frame. This can therefore predict the full probability distribution for the number of cell divisions simply from the average properties (under the simplest assumption of a single population with a set division rate). Interestingly, we found that this simple assumption was enough to reproduce the global features of our experimental dataset (S3B Fig).

Secondly, starting from a single labelled cell undergoing stochastic fate choices, the clone size distribution is expected to rapidly converge towards simple universal scaling laws [79]. In particular, for a 3D organ, the cumulative probability of clones consisting of $k$ cells $P(k)$ is predicted to adopt a simple exponential distribution at all time points: $P(k) = e^{-\frac{k}{<k>}}$, where $<k>$ is the average clone size at this time point. This can be tested experimentally in datasets by plotting the clone size distribution in semi-log plot (S3A Fig), in which an exponential distribution becomes a straight line. Interestingly, clone size distributions arising from manual reconstructions in the liver were shown to follow closely this trend.

To test this more systematically, we computationally reconstructed clones, while assessing from a statistical perspective the quality of the reconstruction. The core idea of this method [37] is to make use of the different colours in multispectral lineage tracing. In the simplest ideal scenario where all 4 colours are labelled with ¼ proportion each, we can then group clones according to specific rules (e.g., group all cells that are each with a certain distance d of each other in a colour-blind manner) and see whether we would have done "mistakes" under this criteria by grouping cells of different colours together in a clone. Said differently, if a specific method grouped cells of different colours together $x$% of the time, then the probability of grouping cells from 2 different clones of the same colour is $\frac{x}{3}$%.

We first assessed how variable were clonal inductions across different livers. Our dataset consisted of 97 fully reconstructed livers, each with the 3D coordinates (x,y,z) of cells of each of the 4 colours. We first asked what the distribution of total number of labelled cells (of all colours) as per given liver and found that this was highly heterogeneous (S3C Fig): Half of the livers have less than 30 labelled cells, while 10% of livers have more than 100 cells (these long tails

were markedly nonexponential and thus unlikely to occur by the stochasticity of cell division patterns, instead likely due to variable induction frequency).

We then computed for each labelled cell 2 metrics: the probability of finding at a distance r a cell of the same colour $P_{same}(r)$ and of a different colour $P_{diff}(r)$. In an ideal, sparse labelled scenario, cells should be all of the same colour in a certain critical radius $r_{crit}$ (where $P_{same}(r)$ is large and $P_{diff}(r)$ is small), while the probabilities should become identical at larger distance (as the colours of neighbouring clones are uncorrelated). We then calculated these distributions for different subsets of livers (the ones with less than 10 labelled cells, 20, 30, etc.), with the expectation that livers with less labelled cells will show stronger clonality (S3D Fig).

Indeed, for livers with less than 10 labelled cells, we found that $P_{same}(r)$ and $P_{diff}(r)$ showed highly distinct peaks, with cells of a same colour being found less that 50 μm apart and of different colours above. Although these livers are likely to be biased towards small clones, this gives us an indication that around 50 μm distance between cells might be a good criterion for clonal groupings. We further found that for livers with less than 50 cells, we could still find a clear distinction between the peaks of the 2 distributions, while this became less and less obvious when adding more densely labelled livers (S3E Fig). Interestingly, when grouping all cells of a given colour that were within 50 μm of one another, and for livers with less than 40 cells, we found exponential clone size distributions close to the ones from the manual assessment of clonality, with average clone size of 4 to 5 cells.

## Assessment of the FRaeppli transgene recombination

To assess the recombination of the *fraeppli-nls* locus, we followed the same PCR-based strategy to amplify recombined genomic transgene regions as previously described [32]. Heat shock at 26 hpf was performed on *tg(fraeppli-nls, prox1a:kalTA4; hsp70l:phiC31,he1a:lyn:Citrine)* embryos. DNA was extracted from individual embryos at 33, 38, and 55 hpf, and 26 hpf controls without heat shock. The following primers (S1 Table) were used: 666 and 667 (detecting a common region in the transgene), 170 and 662 (detecting TagBFP in recombined transgene), 170 and 346 (detecting mTFP1 in recombined transgene), 170 and 665 (detecting mKate2 in recombined transgene), and 170 and 663 (detecting E2-Orange in recombined transgene). The same PCR reactions were performed on 1 ng of FRaeppli 2.0 plasmid and genomic DNA of 5 dpf recombined, FRaeppli fluorescent protein expressing embryos served as positive controls. For negative controls, the PCR were performed on 1 ng of pre-recombined FRaeppli 1.0 plasmid and genomic DNA of non-heat shocked control embryos at 5 dpf. To detect low amounts of recombined DNA, we used the KAPA2G Fast Genotyping Mix (Merck) for the PCRs.

## Statistics and reproducibility

Statistical analysis was performed using GraphPad Prism (version 9.3.1) or MATLAB 2022a. The two-tailed Student $t$ test; n.s. $p \geq 0.05$ was used to calculate statistical significance. All data are presented as mean ± SD. Unless indicated otherwise, n refers to sample size (e.g., individual embryos) and N refers to biological replicates.

## Supporting information

**S1 Data. Numerical values for all datasets.**
(XLSX)

**S1 Table. Primer sequences.**
(DOCX)

**S1 Raw Images. Uncropped gel images of S2E and Uncropped gel images of S2F Fig.** (PDF)

**S1 Fig. Distinct BEC and hepatocyte proportions: predictive in silico modelling of development and in vivo cell type quantification of postembryonic stages.** (**A**-**C**) Mathematical models simulating hepatoblast differentiation, based on heterogeneous hepatoblast potentials (**A**, **B**) or differential proliferation times (**C**; $n$ = 10). (**D**, **F**) Presentation of 10 μm sections from juvenile (**D**) and adult (**F**) livers stained for *fabp10a:GFP* (hepatocytes), *tp1:H2B-mCherry* (BECs), and DAPI (nuclei). (**E**) Relative distribution of BECs and hepatocytes in juvenile liver ($N$ = 4, $n$ = 4 livers and 18 ROIs). (**G**) Relative distribution of BECs and hepatocytes at the organ centre ($N$ = 1, $n$ = 1 liver, 4 sections) or periphery in adult livers ($N$ = 1, $n$ = 1 whole-mount liver). The numerical values that were used to generate the graphs in (**A**-**C**, **E**, **G**) can be found in S1 Data.
(TIF)

**S2 Fig. Defining parameters for lineage tracing experiments using the FRaeppli-NLS system.** (**A**, **B**) (B) A 5 μm projection of *tg(prox1a:kalTA4; UAS:GFP)* embryos stained for 2F11 and DAPI at 80 hpf (**A**) and 96 hpf (**B**). White arrowheads indicate GFP$^+$ BEC nuclei, and yellow arrows highlight GFP$^-$ endothelial cells. ($N$ = 2, $n$ = 8 livers). (**C**) A 10 μm projection of an adult liver section stained for Prox1 (magenta) and Anxa4 (green), white arrowheads indicate Prox1$^+$ BEC nuclei, and yellow arrows highlight Prox1$^-$ endothelial cells. The Prox1 signal was filtered using a median filter with a 3-pixel kernel for better visualisation. ($N$ = 2, $n$ = 6 sections). (**D**) Schematic representation of the stepwise activation times of the *fraeppli* transgene. (**E**) PCR amplification of the mKate2 locus in individual embryos at 26 hpf or 33 hpf upon heat shock–mediated recombination at 26 hpf. ($N$ = 2, $n \geq$ 16 embryos). (**F**) Distribution of the number of FRaeppli recombined loci per embryo upon heat shock at 26 hpf determined by PCR amplification of the recombined transgene. Band intensities at 26 hpf were about 4–6 times lower compared to later time points ($N$ = 2, $n \geq$ 11 embryos). (**G**) Quantification of total liver cell numbers, encompassing hepatocytes and BECs, during development ($N$ = 4, $n \geq$ 12 livers). Different shape data points indicate different experiments. (**H**, **I**) *fraeppli-nls* embryo activated by *phiC31* mRNA injection showing only TagBFP and mTFP1 expression at 60 hpf (**H**), and expression of all 4 FRaeppli FPs at 120 hpf ($n$ = 4 livers) (**I**). Temporal FP colour detection reflects the individual protein maturation times and depends on the strength of the respective Gal4-driver [32]. (**J**) Timelapse of TagBFP$^+$ and mTFP1$^+$ cells using spectral imaging of the liver upon heat shock induction at 9 hpf ($N$ = 2, $n$ = 3 livers). Some neighbouring cells stay close together (magenta arrow), while others move up to 20 μm apart (green arrows). (**K**, **L**) *fraeppli-nls* embryo reimaged at 60 hpf, 72 hpf, and 100 hpf. In sparse recombined embryos (**K**), not all 4 FRaeppli colours are expressed at 100 hpf and an individual labelled cell divides 2 times ($n$ = 2 livers). In highly recombined embryos expression of all 4 colours is visible at 100 hpf ($n$ = 10 livers). (Total $N$ = 2, $n$ = 18 livers). The numerical values that were used to generate the graphs in (**F**, G) can be found in S1 Data.
(TIF)

**S3 Fig. Clonal analysis using the *fraeppli-nls* system.** (**A**) Schematic representation of the crossing scheme of *fraeppli-nls* and *hsp70l:phiC31; prox1a:kalTA4* fish. Exogenous PhiC31 integrase, expressed upon immediate temperature change of the medium, recombines attB/attP sites in the FRaeppli FP colour cassette. KalT4 drives the expression of the FRaeppli FPs. (**B**) Schematic showing the assignment criteria for manual clone definition based on distances between labelled cells of the same colour. (**C-F**) Intercellular distance within mixed clones (**C**), pure hepatocyte clones (**D**, **E**), or pure BEC clones (**F**); (**C-F**) correspond to examples shown

in Fig 3D–3F. (**G**) Pure BEC clone, assignment based on the elongated cell shape.
(TIF)

**S4 Fig. Proportions of mixed clones and quantitative assessment of clonality.** (**A**) Distribution of all labelled cells per livers. (**B**) Frequency of manually assigned mixed clone sizes ($N =$ 5, $n = 16$ clones). (**C**) Cell type distribution within a mixed clone ($N = 5$, $n = 16$ clones). (**D**) Clone size distribution for different colours represented in a semi-log plot shows highly consistent values and good fit to a simple exponential distribution (black line), expected for a single population undergoing stochastic division. (**E**) Number of cell divisions of manually defined clones fit a Poisson distribution (black line) as expected for stochastic divisions. (**F**) Cumulative probability of a certain number of labelled cells per liver ($N = 6$, $n = 97$), showing that most livers have less than 50 labelled cells, but with heavy tails (10%–20%) of highly induced livers. (**G**) Probability that a given cell had a neighbouring cell with the same (red line) or a different colour (black line). Plots show subsets of the data that included livers with a total number of less than 10, 40, 50, or 100 labelled cells. Both distributions show high overlap for highly induced livers, which signifies poor clonality. For distances of less than 50 μm and livers with less than 40 cells, the ratio of "same" to "different" colour is high, meaning that nearby cells of the same colour are unlikely to be nonclonal. (**H**) Manually determined size distribution of clones (black line) plotted together with different reconstructed clone size distributions. Different lines correspond to the regrouping of neighbouring cells of the same colour in the same clone if present within defined radii. Plots show subsets of the data that included livers with a total number of less than 10, 40, 50, or 100 labelled cells. The numerical values that were used to generate the graphs can be found in S1 Data.
(TIF)

**S5 Fig. Lineage tracing reveals a heterogeneous contribution of hepatocytes during postembryonic growth.** (**A**) Adult liver displaying clones along the central vein; recombination was induced in hepatoblasts at 26 hpf ($n = 9$ livers). (**B, C**) Adult livers exhibiting giant clusters in the ventral lobe ($n = 3$ livers). For (**A-C**) total numbers: $N = 9$, $n = 79$ livers. (**D**) Adult liver with a cluster along a central vein ($n = 5$ livers) and (**E**) clusters oriented in lateral stripes ($n = 10$ livers) upon recombination induced in hepatocytes. For (**D, E**) total numbers $N = 4$, $n = 31$ livers). (**F**) Giant clusters in the ventral lobe are also apparent in juvenile livers when labelling was induced at 4 dpf in hepatocytes ($n = 1$; total $N = 5$, $n = 42$ livers). (**G**) Confocal section showing the *kdrl*:*GFP*+ sinusoidal architecture in the adult liver counterstained with DAPI ($N = 1$, $n = 2$ livers). (**H-J**) No recombined cells were detected in 84% noninduced control livers of long-term lineage tracing experiments showed no recombined cells (**H**; $N = 15$, $n = 84$ livers), and the majority of recombined samples ($N = 15$, $n = 100$) show only one recombined clone of a few labelled cells (**I, J**; $N = 15$, $n = 16$ livers).
(TIF)

**S6 Fig. Peripheral growth during liver development.** (**A, B**) Distribution of nuclear distance to the liver surface displayed for hepatocytes and EdU+ hepatocyte (**A**), and BECs and EdU+ BECs (**B**) ($N = 2$, $n \geq 6$ livers). (**C, D**) Distribution of nuclear distance to the nearest neighbour (NN) shown for hepatocytes and EdU+ hepatocytes (**C**) and BECs and EdU+ BECs (**D**) ($N = 2$, $n \geq 8$ livers). The numerical values that were used to generate the graphs can be found in S1 Data.
(TIF)

**S7 Fig. Hepatic growth dynamics of postembryonic zebrafish.** (**A**) Fish standard length (SL) plotted against fish age. (**B, C**) Fish weight (**B**) and liver weight (**C**) increases with SL represented in a semi-log plot. (**D**) Liver-to-body weight ratio during postembryonic growth is

constant in adult fish. ($N > 10$, $n \geq 300$ fish). Gender of the corresponding samples is colour coded: male (blue), female (pink), and ND (green). The numerical values that were used to generate the graphs can be found in S1 Data.
(TIF)

**S8 Fig. Postembryonic ventral lobe formation.** (A, B) Confocal images of the same liver showing the embryonic left liver lobe at 5 dpf with a 3-cell mKate2[+] clone (**A**) and at juvenile stage (SL = 14.4 mm) including a continuous Kate2+ clone in the ventral lobe ($N = 1$, $n = 1$ liver). (**C, D**) Juvenile livers (C–SL = 8.46 mm and D–SL = 10.93 mm) with connected clusters that are oriented along the tissue edge and spread through the left and the ventral lobe. Arrows indicate cluster growth direction ($N = 4$, $n = 14$ livers). (**E-P**) Brightfield images of stages I-VI livers in loco within the fish (**E, G, I, K, M, O**) or dissected out (**F, H, J, L, N, P**). In (**M**), the liver is removed and the gut bend is visible. A, anterior; P, posterior; R, right; L, left; RL, right lobe; LL, left lobe; VL, ventral lobe.
(TIF)

## Acknowledgments

We thank the Ober group for discussion and comments on the manuscript. We are grateful to Dr. F. Lemaigre for feedback on the manuscript and Dr. T. Piotrowski for invaluable support. We thank the department of experimental medicine (AEM) in Copenhagen for expert fish care. We gratefully acknowledge the DanStem Imaging Platform (University of Copenhagen) for support and assistance in this work.

## Author Contributions

**Conceptualization:** Elke A. Ober.

**Formal analysis:** Iris. A. Unterweger, Edouard Hannezo, Ala Trusina.

**Funding acquisition:** Ala Trusina, Elke A. Ober.

**Investigation:** Iris. A. Unterweger, Julie Klepstad, Edouard Hannezo.

**Methodology:** Iris. A. Unterweger.

**Project administration:** Pia R. Lundegaard, Elke A. Ober.

**Software:** Julie Klepstad, Ala Trusina.

**Supervision:** Elke A. Ober.

**Validation:** Iris. A. Unterweger, Edouard Hannezo.

**Visualization:** Iris. A. Unterweger.

**Writing – original draft:** Iris. A. Unterweger.

**Writing – review & editing:** Iris. A. Unterweger, Julie Klepstad, Edouard Hannezo, Pia R. Lundegaard, Ala Trusina, Elke A. Ober.

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
