## [Editor Report · Decision Letter 0]

27 Jan 2023

Dear Dr Ober, 

Thank you for submitting your manuscript entitled "Lineage tracing identifies heterogeneous hepatoblast contribution to cell lineages and postembryonic organ growth dynamics" for consideration as a Research Article by PLOS Biology.

Your manuscript has now been evaluated by the PLOS Biology editorial staff as well as by an academic editor with relevant expertise and I am writing to let you know that we would like to send your submission out for external peer review.

Once your full submission is complete, your paper will undergo a series of checks in preparation for peer review. After your manuscript has passed the checks it will be sent out for review. To provide the metadata for your submission, please Login to Editorial Manager (https://www.editorialmanager.com/pbiology) within two working days, i.e. by Jan 31 2023 11:59PM.

Kind regards,

Luke

Lucas Smith, Ph.D.

Associate Editor

PLOS Biology

lsmith@plos.org

---

## [Decision Letter · Decision Letter 1]

15 Mar 2023

Dear Dr Ober,

Thank you for your patience while your manuscript "Lineage tracing identifies heterogeneous hepatoblast contribution to cell lineages and postembryonic organ growth dynamics" was peer-reviewed at PLOS Biology. It has now been evaluated by the PLOS Biology editors, an Academic Editor with relevant expertise, and by several independent reviewers. 

In light of the reviews, which you will find at the end of this email, we would like to invite you to revise the work to thoroughly address the reviewers' reports.

Overall, the reviewers appreciate the detailed dataset provided here and note that it will open up interesting lines of study. While Reviewers 1 and 2 have requested fairly minor changes, Reviewer 3 has raised more substantial concerns including that a more complete description of the system used here is needed, that more work is necessary to support the most provocative conclusion of the study, and that the manuscript could be revised to be more focused. We think the reviewer requests would need to be carefully addressed before we can consider your manuscript for publication.

Given the extent of revision needed, we cannot make a decision about publication until we have seen the revised manuscript and your response to the reviewers' comments. Your revised manuscript is likely to be sent for further evaluation by all or a subset of the reviewers.

**IMPORTANT - SUBMITTING YOUR REVISION**

*Re-submission Checklist*

*Published Peer Review*

*PLOS Data Policy*

*Blot and Gel Data Policy*

Sincerely,

Lucas

Lucas Smith, Ph.D.

Associate Editor

PLOS Biology

lsmith@plos.org

REVIEWS:

Reviewer #1: This article by Unterweger et al studies the cellular origin and growth of the liver in the zebrafish.

It combines mathematical models with genetic fate mapping and 3D organ imaging at single cell level followed by sophisticated statistical analysis to understand to which extent progenitor cells contribute to hepatocyte and biliary endothelial cell (BEC) generation.

If there are or not multipotent, bi- or uni- potent hepatocyte and BEC has been debated for very long. This clonal analysis using their recently published fraeppli-nls line combined with reporters and recombination lines specific for BECs and live progenitor cells respectively reveals new evidence on this question. The authors find that the liver is build from bi and unipotent progenitors. Their data suggests a ratio of 80% unipotent hepatocyte-producing hepatoblasts and 20% bipotent ones. The authors observe that clonal expansion associated to endothelial sinusoids and do not observe a clear pattern in the left and right lobes that would suggest that liver growth and shape is driven by directed proliferation. They also show that clonal expansion is probably stochastic, meaning that there is not a particular population with a higher degree of expansion. They do however observe that the ventral lobe has a different growth pattern with large clones found herein, what could indicate that few progenitors massively expands and give rise to this lobe. Further, they observe very unexpectedly, that hepatocytes are polyploid transiently during growth, but in the adult polyploidy is lost. 

The characterization of liver development absolutely exceptional. The author have performed an incredibly detailed and meticulous work, that will be of great use no only to understand cell fate determination of liver progenitors but also as an extraordinary helpful morphological atlas of zebrafish liver formation from the embryo to juveniles until adults.

Obviously, this nice work calls immediately for more questions:

How is the clone distribution in a pathological situation or injury scenario? What genetic programs make the cells decide to become a hepatocyte or a BEC? What happens to the polyploid hepatocytes?

 However, these questions seem to go beyond the current article, that already counts with 8 main and 7 Supplementary Figures.

Minor comment: 

* Making the data available might be useful for other researchers to analyse further parameters. Currently there is no statement on data availability.

* Page 10. Recombined liver clusters not livers

Reviewer #2: In this study, Ober and colleagues used mathematical modeling, quantitative biology, and multispectral lineage tracing to investigate the contribution of individual hepatoblasts to hepatic lineage formation and organ growth in the juvenile and adult fish. Such interdisciplinary approach unveiled many interesting features and novel mechanisms underlying liver organogenesis. First, they reported a constant ratio between the two main cell types in the liver, the hepatocytes and biliary epithelial cells, which is established at early embryonic stages and maintained throughout development, growth, and homeostasis in zebrafish. This ratio is the result of a heterogeneous lineage contribution of unipotent (the majority) and bipotent hepatoblasts to the liver, which is against the dogma of all hepatoblasts being bipotent. Then, by extending their lineage tracing analysis to adulthood, the authors discovered that embryonic cells contribute heterogeneously to postembryonic liver growth. Importantly, the findings reported in this manuscript suggest that both heterogenous lineage contribution and heterogenous proliferation dynamics during post-embryonic growth are not an intrinsic property of individual progenitors but more likely due to external inductive signals and surrounding microenvironmental niches (e.g., periphery, vessels, mesothelium).

Overall, I think this a strong paper and perfectly suited for PloS Biology. The results are compelling, and the methodology innovative and powerful. The manuscript is well-written and figures presented are of high quality. The findings are extremely interesting, open up many new avenues of investigation in the field of liver biology (development & regeneration) and might have an impact on the development of future cell therapies for curing liver diseases, once conservation in mammalian species is assessed. 

Minor points that could be addressed in the discussion: 

- It would be important to discuss any contribution of cell death and/or competition to the heterogenous differentiation and expansion of progenitors during early liver development.

- The authors used the prox1 transgene as driver in the FRaeppli multicolour lineage tracing experiments. Can they exclude that the transgene labelling efficiency might be a confounding factor in the lineage tracing results?

Reviewer #3: 

This is a review of MS entitled "Lineage tracing identifies heterogenous hepatoblast contribution to cell lineages and postembryonic organ growth dynamics" by Unterwegar et al. In this MS the authors use a newly developed (for zebrafish) lineage tracing mechanism termed FRaeppli-NLS. This system makes use of infrequent recombinase-mediated activation of one of 4 fluorescent proteins to label and follow clones throughout development and into adulthood. Remarkably the non-overlapping spectra of the 4 FPs allow for reliable fate mapping and is even more powerful when combined with transgenic lineage markers and whole organ confocal analysis as performed herein. The authors aim to exploit this system, in combination with mathematical modeling, to understand hepatoblast potency during development and postnatally. Their models are produced by varying hepatoblast potency and proliferation in order to attain the 1:9 BEC :hepatoblast ratio observed at 120 hpf. BEC and hepatocyte markers are used to experimentally determine that the 1:9 ratio is established at the onset of cell commitment @ 48 hpf. Edu labelling is used to identify proliferation rates and find a slight increase in BEC proliferation between 48-120 hpf, which is deemed too similar to hepatocytes to be significant. The FRaeppli system is used to assess hepatoblast potency by label clones 9-12 hours prior to commitment at 31-35 hpf, and imaging the resulting livers 100 hpf. Surprisingly they find that a minority of the hepatoblast clones are bipotential, while the majority gave rise solely to hepatocytes. Most of the clones produced fewer cells than expected based on their initial Edu experiments. They use a similar experimental strategy to label clones prior to specification and then allow their embryos to develop through juvenile and adult stages to examine how labeled clones contribute to the adult organ. They produce novel information demonstrating that hepatocytes clones develop along central veins, or medial-lateral strives of hepatocytes and demonstrate, by imaging clones at 120 hpf and tracing over months, how the ventral lobe is formed. They find that most proliferation occurs from cells closest to the cell surface, demonstrating that the significant growth observed between 120 hpf through 3 mpf is spurred by dividing cells at the periphery of the organ and that polyploidy is not a mechanism utilized significantly by zebrafish during normal growth. 

Overall this paper uses a novel and promising labelling system and the resultant confocal images are very detailed and aid in understanding how the liver is expands from progenitors. The most convincing data is that which describes how the liver develops post-embryonically. This data is mainly descriptive, and lacks a coherent theme, but is some of the first data produced from the zebrafish model. The embryonic data on hepatoblast potency is the most provocative and important however because this data is reliant on the reproducibility and defined occurrence of clone production there are several parameter gaps that must be sufficiently addressed. Furthermore, more attention should be paid to explaining the FRaeppli system, so that the data is accessible to a broader audience. Finally, introduction and discussion could be better focused as the logic in both is often difficult to follow. 

Major weaknesses (in order of appearance in the MS, not of importance)

A) p. 3 last few sentences on the page. This part of the introduction is confusing. Ref 13: the authors methodically demonstrate that single cells are recombined at E8.25, coincident with hepatic specification. Demonstrating that the earliest hepatoblasts are bipotent. Ref12: demonstrates some E10.0 hepatoblasts are unipotent while most are bipotent. Couldn't this be seen as a gradual restriction of hepatoblast fate over time? The end of this paragraph is not focused and warrants more attention. 

B) Fig 1B' Are the grey cells the HNF4a + cells? What is the difference between the grey cells and those with a white core?

C) Fig 2: An image used for cell proliferation should be included in Figure 2.

D) Fig 2: What do the shapes in each category represent? How many livers were examined at each stage? Why are there fewer shapes at 144hpf? 

E) Fig 2: It appears as if the trend is that BECs differentiate faster at all time points (with significance at 96 and 120hpf). This point is not mentioned in the text other than "minimal differences between BECs and hepatocyte proliferation rates" are observed. Please describe how such changes might affect the model and if others have observed such differences in proliferation rates.

F) p. 8: A more complete description of the Fraeppli-NLS system is needed for the reader to critically evaluate this experimental system. For example- what are the dynamics of the detection? How soon after recombination can the clones be visualized? How many cells are in each clone at the onset of detection? Most importantly- how long does it take for all clones to be established? Do the number of clones increase after differentiation at 48 hours? If so, then could the small number of cells in a clone reflect a developmentally later onset of recombination? This could explain the higher number of smaller than expected clones.

G) p.8 In the 2022 Caviglia et al paper defining this method, Cre is used to induce recombination of the fluorescent reporter. It is not clear to this reviewer how Cre is induced in the system used in the current manuscript. 

H) Fig 3D: This image is not convincing and appears to be two different clones, each clone labeled with the same color, one differentiated toward hepatocyte the other toward BEC. It is also striking that there are similarly large numbers of cells in each lineage. In this example the hepatocytes and BECs are in discreet positions of the liver and each lineage is clustered together. Do all bipotent cells display a similar spatial organization (in distinct portions of the liver)? 

I) Fig3: Is there a difference in the number of cells in clones that display a unipotent hepatoblast fate and those that display a bipotent hepatoblast/BEC fate? 

J) Fig 3. The distribution of clone types (represented by D,E,F) within the 85 labelled livers should be provided. For example- this could demonstrate that hepatoblast-only clones are found throughout most of the labeled livers or are restricted mainly to those livers that do not have bipotent clones. Given that the number of cells produced by any one clone typically did not undergo as many cell divisions as predicted, is it possible that the cells were labeled later than expected?

K) Page 9: paragraph 2. The reason that the hepatocyte clones were focused on rather than BEC clones is confusing.

Minor Edits

A) Fig 3B: Why does the timeline extend to 120+ hours? The paper states that embryos are fixed and imaged at 100 hpf. Furthermore this timeline is confusing when contrasted with Fig 5 that states time of imaging and time of fixation. 

B) Fig3. The legend for D-F states that N=6. What does the N represent? 

C) Figure 5: How many total livers were examined? What do the N and n represent in this figure legend?

D) Discussion: page 16. The second full sentence is not complete. 

E) Page 17: The sentence beginning "the overproportionate contribution…" is difficult to comprehend.

F) Page 17: Final paragraph " We propose that both lineage and proliferation heterogeneity is not intrinsic hepatoblast property…" Insert "an" before "intrinsic".

---

## [Decision Letter · Decision Letter 2]

3 Aug 2023

Dear Dr Ober,

Thank you for your patience while we considered your revised manuscript "Lineage tracing identifies heterogeneous hepatoblast contribution to cell lineages and postembryonic organ growth dynamics" for publication as a Research Article at PLOS Biology. Your revised study has now been evaluated by the PLOS Biology editors, the Academic Editor and two of the original reviewers. 

The reviews are appended below. As you will see, both reviewers agree that the manuscript has been strengthened, but Reviewer 3 has a number of lingering concerns that we think should be addressed in another revision that we anticipate should not take you very long. We will then assess your revised manuscript and your response to the reviewers' comments with our Academic Editor aiming to avoid further rounds of peer-review, although might need to consult with the reviewers, depending on the nature of the revisions.

**IMPORTANT: As you address these last reviewer comments, please also attend to the following editorial requests: 

1) In the ethics statement, in your materials and methods section, please provide the approval number for the protocol approved by Department of Experimental Medicine (AEM) of the University of Copenhagen.

2) Thank you for providing the data underlying your figures in the supporting information. Please reference this dataset in each of your figure legends. 

3) Per journal policy, as the code that you have generated is important to support the conclusions of your manuscript, we require that you make it available without restrictions upon publication. Please ensure that the code is sufficiently well documented and reusable, and that your Data Statement in the Editorial Manager submission system accurately describes where your code can be found (ideally this will be made available on a repository or github).

**IMPORTANT - SUBMITTING YOUR REVISION**

*Resubmission Checklist*

*Published Peer Review*

Sincerely,

Luke

Lucas Smith, Ph.D.

Senior Editor

PLOS Biology

lsmith@plos.org

REVIEWS:

Reviewer #2: In the revised version of this manuscript the authors have satisfactorily responded to all the points raised by this referee. 

The text revision and new set of experiments included has greatly improved the work.

Reviewer #3, Kimberly D Tremblay (note, reviewer 3 has signed this review): PBIOLOGY-D-23-00163 Review #2 

I appreciate the authors detailed response to many of my previous critiques. Several addressable yet important concerns were not answered satisfactorily.

1) The response to query E regarding Figure 2: 

Figure 2E clearly demonstrates that BECs exhibit significantly higher proliferation rate than hepatocytes at 96 and 120 hours. Line 154 states "our results reveal minimal differences between BEC and hepatocyte proliferation rates suggesting progenitor potential as a major factor for the establishment of the 1:9 BEC to hepatocyte ratio in vivo..". Please explain how this proliferation data reveals "minimal differences in proliferation rates". If as explained in your rebuttal that Edu does not represent proliferation rate, then explicitly remark on how the data in figure 2 supports minimal differences between BEC and hepatocyte proliferation rates. If these results are incongruous with the proposed models explain how.

2) In response to query F: The explanation provided in the rebuttal regarding the temporal differences in clone detection due to differences in maturation of the distinct fluorescent cassettes used should be stated in the MS. This allows the reader to accurately understand the dynamics of the labeling system. 

3) In response to question posed in query F (and in query J): "Could the higher than expected number of small clones reflect a later than expected onset of recombination?" This question has not been answered. I am not referring to the number of clones observed in non-heat shocked control animals (which are referred inaccurately in the MS as "unspecific recombination events", unspecific is not an appropriate word), although this is an essential parameter that should have originally been included in the MS. What are the data supporting the stated maturation of PhiC31 for the 4 hours after heat shock (26-30 hpf) and the attB/P recombination confined to the 3 hours after that (30-33 hpf). Where are the experimental details that support this narrow time-frame? Is it possible that recombination occurs later than this because of delayed PhiC31 maturation or because of a longer than expected life of the attB/P recombinase? 

4) In response to query F and J: The revised text inserted in lines 250-253 is not stated in a clear manner (and in response to the new comment 3 may still need to be updated). "An overrepresentation of small clones caused by later than expected labelling events represents a formal possibility, however, less likely given that PhiC31 recombination occurs fast and efficient (Figs S2E,F) and low unspecific recombination (1.8%) observed in controls." Instead of "Recombination occurs fast and efficient" perhaps recombination occurs quickly and efficiently". Furthermore, "Unspecific" is not an appropriate word to use (perhaps you mean non-specific?). 

5) In response to query G: In Figure S2 it appears as if exogenous PhiC31 mRNA injection is used in H and I. However you state in your rebuttal that only exogenous phiC31 under the control of hsp70l promoter is used. Please ensure that the details for such exogenous mRNA injection are included in the methods and are overtly stated in the legend for Fig S2. 

6) The revisions provided in the new text on lines 179-199 needs to be better edited for accuracy.

Line 180 "The spectra…" I think that a more accurate way of stating what you mean is: " Thes spectra of FRaeppli FPs does not overlap with the spectra of EGFP allowing for the simulataneous use of the transgenic tp1:EGFP…"

Line 181-183 "To ensure… we used prox1 as a driver" insert "as a UAS driver"

Line 186-188 "For conditional induction of recombination…for conditional induction of cell recombination". Please make this sentence more accurate and concise.

Lines 188-191: As per the timing question posed in Query 4 above, the sentence beginning with "PhiC21 maturation and subsequent attB/P recombination….takes about 6-7hours (note: insert space between 7 and hours) and thus occurs prior to fate commitment" unless evidence excludes later recombination occurring as a result of PhiC21 the statement should be "thus initiates prior to fate commitment".

---

## [Editor Report · Decision Letter 3]

29 Aug 2023

Dear Dr Ober,

Thank you for the submission of your revised Research Article "Lineage tracing identifies heterogeneous hepatoblast contribution to cell lineages and postembryonic organ growth dynamics" for publication in PLOS Biology. Your manuscript has been assessed by the PLOS Biology editorial team and the Academic Editor, and we are satisfied by the changes made in response to the last editorial and reviewer requests. Therefore, on behalf of my colleagues and the Academic Editor, Marianne E. Bronner, I am pleased to say that we can in principle accept your manuscript for publication, provided you address any remaining formatting and reporting issues. These will be detailed in an email you should receive within 2-3 business days from our colleagues in the journal operations team; no action is required from you until then. Please note that we will not be able to formally accept your manuscript and schedule it for publication until you have completed any requested changes.

PRESS

We frequently collaborate with press offices. If your institution or institutions have a press office, please notify them about your upcoming paper at this point, to enable them to help maximize its impact. If the press office is planning to promote your findings, we would be grateful if they could coordinate with biologypress@plos.org. If you have previously opted in to the early version process, we ask that you notify us immediately of any press plans so that we may opt out on your behalf.

Sincerely, 

Lucas Smith, Ph.D.

Senior Editor

PLOS Biology

lsmith@plos.org